# Seminal fluid compromises visual perception in honeybee queens reducing their survival during additional mating flights

Joanito Liberti[1†‡*], Julia Görner[2], Mat Welch[2], Ryan Dosselli[2,3], Morten Schiøtt[1], Yuri Ogawa[4], Ian Castleden[2], Jan M Hemmi[4], Barbara Baer-Imhoof[5], Jacobus J Boomsma[1*], Boris Baer[5*]

[1]Centre for Social Evolution, Department of Biology, University of Copenhagen, Copenhagen, Denmark; [2]ARC Centre of Excellence in Plant Energy Biology, The University of Western Australia, Crawley, Australia; [3]Centre for Evolutionary Biology, School of Biological Sciences, The University of Western Australia, Crawley, Australia; [4]School of Animal Biology and UWA Oceans Institute, The University of Western Australia, Crawley, Australia; [5]Centre for Integrative Bee Research (CIBER), Department of Entomology, University of California, Riverside, Riverside, United States

**\*For correspondence:**
joanito.liberti@unil.ch (JL);
jjboomsma@bio.ku.dk (JJB);
boris.bar@ucr.edu (BB)

**Present address:** [†]Department of Fundamental Microbiology, University of Lausanne, Lausanne, Switzerland; [‡]Department of Ecology and Evolution, University of Lausanne, Lausanne, Switzerland

**Competing interests:** The authors declare that no competing interests exist.

**Abstract** Queens of social insects make all mate-choice decisions on a single day, except in honeybees whose queens can conduct mating flights for several days even when already inseminated by a number of drones. Honeybees therefore appear to have a unique, evolutionarily derived form of sexual conflict: a queen's decision to pursue risky additional mating flights is driven by later-life fitness gains from genetically more diverse worker-offspring but reduces paternity shares of the drones she already mated with. We used artificial insemination, RNA-sequencing and electroretinography to show that seminal fluid induces a decline in queen vision by perturbing the phototransduction pathway within 24–48 hr. Follow up field trials revealed that queens receiving seminal fluid flew two days earlier than sister queens inseminated with saline, and failed more often to return. These findings are consistent with seminal fluid components manipulating queen eyesight to reduce queen promiscuity across mating flights.
DOI: https://doi.org/10.7554/eLife.45009.001

## Introduction

Seminal fluid is a complex mixture of proteins and metabolites with multiple functions to enhance male reproductive success (*Poiani, 2006*; *Avila et al., 2011*). It keeps sperm alive and motile, protects against pathogens, and regulates sperm capacitation, the final maturation step that enables sperm to fertilise eggs (*Chapman, 2001*; *Poiani, 2006*; *Otti et al., 2009*). When females mate with multiple males, seminal fluid components can become agents of sexual selection and harm rival ejaculates while others manipulate female physiology to enhance a specific male's reproductive success (*Parker, 1970*; *Birkhead and Møller, 1998*; *Chapman et al., 2003a*; *den Boer et al., 2010*). These interactions have been documented in detail in the fruit fly *Drosophila melanogaster* where seminal fluid promotes fast oviposition and reduces the willingness of females to seek additional copulations (*Chen et al., 1988*; *Liu and Kubli, 2003*; *Chapman et al., 2003b*). A key molecule responsible for these effects is the sex peptide, a seminal fluid peptide that crosses the vaginal wall to enter the hemolymph and bind to a G-protein-coupled receptor on neurons that activate a signaling

**eLife digest** For social insects like honeybees it is beneficial if their queens mate with many males, because genetic diversity can protect the hive against parasites. Early in life, a honeybee queen has a short period of time in which she can fly out to mate with males before returning to the hive with all the sperm needed to last for a lifetime. Queens that have mated on their first flight may embark on additional mating flights over a few consecutive days to further increase genetic variability in their offspring. This is problematic for a male that has already mated because the more males that inseminate the queen the fewer offspring will carry on his specific genes. This results in sexual conflict between males and queens over the number of mating flights.

In many animals, males manipulate females using molecules in seminal fluid to reduce the chances of the female mating again and honeybee males may use a similar strategy. Previous studies revealed that insemination alters the activity of genes related to vision in a honeybee queen's brain. This could be one way for the males to prevent queens from embarking on additional mating flights.

Now, Liberti et al. find support for this idea by showing that seminal fluid can indeed trigger changes in the activity of vision-related genes in the brains of honeybee queens, which in turn reduce a queen's opportunity to complete additional mating flights. Queens inseminated with seminal fluid were less responsive to light compared to queens that were exposed to saline instead. Electronic tracking devices affixed to queens showed that the seminal fluid-exposed queens left for mating flights sooner but were more likely to get lost and to not return to their hives compared to the saline-exposed queens.

The experiments support the idea of a sexual arms race in honeybees. Males use seminal fluid to cause rapid deteriorating vision in queens, thus reducing their likelihood of leaving the hive to mate again and to find males when they do fly again. The queens try to counteract these effects by leaving for mating flights sooner, thereby increasing offspring genetic diversity and the success of their colonies. Further studies will be needed to find out how the honeybee sexual arms race varies across seasons, bee races, and geographic ranges. Such information will be useful for honeybee breeding programs, which rely on queen mating success and hive genetic diversity to ensure hive health.

DOI: https://doi.org/10.7554/eLife.45009.002

transduction cascade (*Yapici et al., 2008*). Similar phenotypic effects have been reported in other insects, but without a detailed understanding of the molecular mechanisms involved (e.g. *Craig, 1967*; *Gillott and Langley, 1981*; *Baer et al., 2001*; *Hayashi and Takami, 2014*). Recent proteomic characterizations of seminal fluid in honeybees revealed a number of proteins with the potential to interact with neurons (*Baer et al., 2009*; *Grassl et al., 2017*), suggesting that the seminal fluid of honeybee males (known as drones) might be able to manipulate queen mating behaviour.

Several decades of research on ants, social bees and social wasps have shown that obligate multiple insemination of queens is always evolutionarily derived from single paternity ancestors (*Hughes et al., 2008*; *Boomsma, 2013*). These studies imply that obligate polyandry evolved predominantly in lineages with large and long-lived colonies where genetically diverse workers make colonies more likely to survive and reproduce (*Mattila and Seeley, 2007*; *Mattila et al., 2012*). Social insect males mating with a focal queen will benefit from shared paternity if that is their only route to reproductive success, but the optimal number of inseminations are expected to be higher for a queen than for the males inseminating her (*Koeniger, 1990*). This form of sexual conflict is unlikely to affect flight behaviour when queens depart as virgins and inseminations follow each other in quick succession. If queens never fly again, it is thus reasonable to assume they will store a genetically diverse fraction of all sperm received to maximise life-time reproductive success (*Jaffé et al., 2012*) while suppressing sperm competition between ejaculates (*Starr, 1984*; *Boomsma et al., 2005*), a sequence of events that is increasingly well documented (*Mattila and Seeley, 2007*; *den Boer et al., 2010*; *Mattila et al., 2012*). Honeybees are the only social insect lineage so far known to deviate from this rule because a significant fraction of queens embark on additional flights on subsequent days when they are no longer virgins (*Woyke, 1964*; *Tan et al., 1999*). Newly mated

honeybee queens return to their hives and complete the process of sperm storage over several days, which corresponds with a physiological transition to become an established egg-laying mother queen (*Woyke, 1983*; *Winston, 1987*). If queens decide during this brief time period to leave for a second or third risky mating flight, their choice will reduce the fitness of drones whose sperm she has already acquired but not yet stored.

Previous studies of the brains, ovaries and fat bodies of *Apis mellifera* queens demonstrated that complex physiological changes occur in response to mating, and that they are controlled by multiple, largely uncorrelated, mechanisms (*Koeniger, 1976*; *Grozinger et al., 2007*; *Kocher et al., 2008*; *Kocher et al., 2009*; *Kocher et al., 2010*; *Niño et al., 2011*; *Vergoz et al., 2012*; *Niño et al., 2013a*; *Niño et al., 2013b*; *Manfredini et al., 2015*; *Brutscher et al., 2019*). For example, both artificial insemination procedures and manipulation of insemination volume trigger ovary activation through mechanical stimuli of stretch receptors in the genital tract that also appear to affect a queen's decision to pursue additional mating flights, independent of the type of substance transferred to the queen's sexual tract (*Kocher et al., 2009*; *Niño et al., 2011*). Queen exposure to carbon dioxide induces similar changes as the ones induced by copulation, for example reducing mating flights number, triggering ovary development and changes in chemical composition of mandibular gland secretion, or altering gene expression in the brain (*Niño et al., 2011*; *Vergoz et al., 2012*; *Niño et al., 2013b*). These studies also found consistently altered expression of genes with known links to immune responses and visual perception of queens after mating (*Grozinger et al., 2007*; *Kocher et al., 2008*; *Kocher et al., 2010*; *Manfredini et al., 2015*). However, the factors inducing these latter effects have remained elusive and the ensuing reductions in visual perception have neither been phenotypically verified nor been interpreted as possible consequences of sexual conflict. Our present study provides an explicit test of the hypothesis that ejaculates contain molecules that affect the neurophysiology and behaviour of queens in such a way that these queens are less likely to acquire additional matings during subsequent mating flights. Unconstrained visual perception by queens is important for a successful return to their hives, but also expected to be essential for locating drone congregation areas during flight. We therefore predicted that seminal fluid compounds could be effective instruments to maximise the fitness interests of drones that already have achieved insemination success, if such compounds target queen photoreception. This conjecture could then be an example of sexual conflict mediated by sensory exploitation, which is known to have created sexual arms-race dynamics in other animals via male-manipulation that induced selection for compensatory traits in females (*Arnqvist, 2006*; *Hollis et al., 2019*).

We used RNA-sequencing to quantify gene expression changes in the brains of queens that we had artificially inseminated with seminal fluid (solely or in combination with sperm) and assessed whether this induced comparable changes in the expression of vision-related genes to those reported for naturally inseminated queens (*Grozinger et al., 2007*; *Kocher et al., 2008*; *Kocher et al., 2010*; *Manfredini et al., 2015*). Because we were able to confirm that the expression of genes involved in phototransduction was indeed altered by seminal fluid, we then phenotypically quantified the visual performance of queens that had been inseminated with seminal fluid with and without sperm, or a saline control fluid. We measured response amplitude and contrast sensitivity of the queens' compound eyes and ocelli over two days after artificial insemination and found that those queens exposed to seminal fluid indeed experienced a significant decrease in visual perception. We completed our study by monitoring natural mating flight activities in an apiary, using queens that had received the same artificial insemination treatments. We found clear effects of seminal fluid on the timing of, and survival during, queen mating flights, consistent with ongoing antagonistic selection for manipulative seminal fluid traits and compensating behavioural defences by queens.

## Results

### Analyses of gene expression in queen brains

We found that 24 hr after queens had received seminal fluid – either pure or as part of ejaculates (i.e. together with sperm) – their brain gene expression profiles were substantially altered compared to queens that had either received mock inseminations (i.e. the entire procedural sequence of artificial insemination but without injecting fluid into their genital tracts) in a first RNA-seq experiment, or

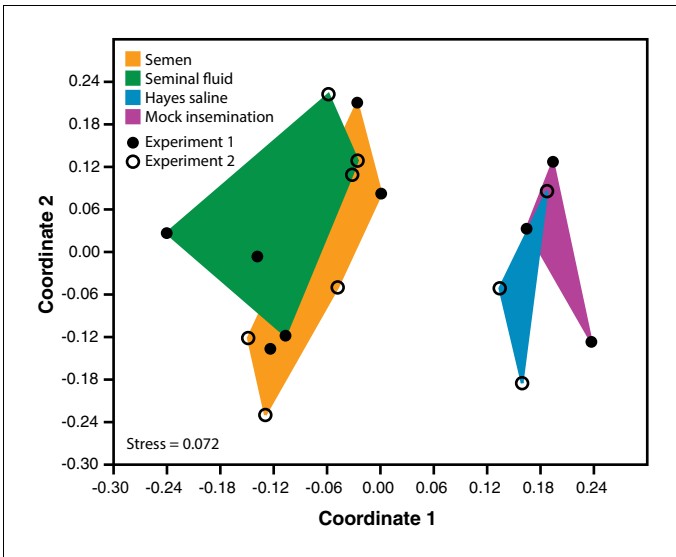

**Figure 1.** Semen and seminal fluid inseminations cause expression changes in honeybee queen brains, as revealed by Non-metric Multidimensional Scaling (NMDS) of Bray-Curtis dissimilarities between brain samples based on the 1327 differentially expressed genes identified with DESeq2 analyses. Samples from truly inseminated queens (semen and pure seminal fluid without sperm) are separated from controls (Hayes saline and mechanical mock insemination) after removal of batch effect caused by two distinct RNA-sequencing experiments being combined. A stress value well below 0.2 of the NMDS analysis indicates that after reduction to two dimensions the plot captures the relevant variation fairly well.

DOI: https://doi.org/10.7554/eLife.45009.003

The following source data and figure supplements are available for figure 1:

**Source data 1.** Table of normalized counts after variance stabilizing transformation, removal of experimental batch effect by removeBatchEffect function in edgeR v.3.12.1 and centring over the mean across samples, used as input for ordination analysis in Paleontological Statistics v.3.04.

DOI: https://doi.org/10.7554/eLife.45009.007

**Figure supplement 1.** Volcano plots presenting significance (-log10 (*P* value)) *versus* log2 (fold change) for all pair-wise comparisons between treatment groups in RNA-sequencing Experiments 1 and 2.

DOI: https://doi.org/10.7554/eLife.45009.004

**Figure supplement 2.** Venn diagram showing overlap between differentially expressed genes identified in each of the pair-wise comparisons of insemination treatments against controls across the two experiments.

DOI: https://doi.org/10.7554/eLife.45009.005

**Figure supplement 3.** Networks representing interactions between enriched GO terms (p<0.05) in all pair-wise comparisons between insemination treatment groups in Experiments 1 and 2.

DOI: https://doi.org/10.7554/eLife.45009.006

control inseminations with Hayes saline only in a second experiment (*Figure 1*). We identified 1327 (8.6% of the honeybee transcriptome) differentially expressed genes (DEGs) across all pair-wise brain-comparisons between treatment groups in the two subsequent RNA-seq experiments, with an over-representation of up-regulated DEGs in queens exposed to pure seminal fluid or semen (i.e. complete ejaculates) (*Figure 1—figure supplement 1* and *Figure 1—figure supplement 2*; see *Supplementary file 1* and *Supplementary file 2* for the number and identity of DEGs identified in each pair-wise comparison, respectively).

Functional enrichment analyses revealed that several Biological Process terms such as 'signal transduction', 'signaling', 'cell communication' and 'response to stimuli' were consistently enriched in all comparisons of semen and pure seminal fluid insemination treatments compared to controls (*Figure 1—figure supplement 3* and *Supplementary file 3*). Enriched GO terms also included 'pathogenesis' (semen vs. mock insemination), 'ocellus pigmentation', 'sleep', 'proteolysis', 'regulation of phagocytosis', 'negative regulation of DNA replication', 'RNA metabolic process' and 'carbohydrate catabolism' (seminal fluid vs. mock insemination), 'lipid catabolic process' (both semen and seminal fluid vs. mock insemination), 'mitotic cell cycle process', 'vitamin transport', 'DNA

packaging', 'catechol-containing compound metabolic process' (semen vs. Hayes saline), 'establishment of epithelial cell polarity' (semen vs. mock insemination, seminal fluid vs. mock insemination and semen vs. Hayes saline), 'negative regulation of MAPK cascade', 'NAD metabolic process' (seminal fluid vs. Hayes saline), 'tryptophan catabolic process to kynurenine' (both seminal fluid vs. mock-insemination and seminal fluid vs. Hayes saline), and several terms related to the transmembrane transport of ions (seminal fluid vs. mock insemination and semen vs. Hayes saline; see *Figure 1—figure supplement 3* and *Supplementary file 3* for more detailed lists, including Molecular Function GO terms).

Although we detected only a single DEG in the semen vs. seminal fluid comparison in the first RNA-seq experiment, the same comparison yielded 802 DEGs in RNA-seq experiment 2, which had greater detection power for unknown reasons. However, all these DEGs had only relatively small expression changes ($-1 <$ log2 (fold change) $< 1$) and most of them were up-regulated in semen compared to seminal fluid (*Figure 1—figure supplement 1*), suggesting that the presence of sperm may have enhanced effects of seminal fluid on the expression of many of these genes. A GO enrichment analysis revealed that these genes were mostly involved in 'signaling', 'cell communication' and 'ion transport' as with seminal fluid alone, but also mediated effects that were only recovered for this semen vs. seminal fluid comparison, such as the Biological Process terms 'ATPase activity' and 'oxidation-reduction process' (*Supplementary file 3*).

We then used GAGE analyses (*Luo et al., 2009*) to identify which signaling and metabolic cascades were altered by the semen and seminal fluid insemination treatments. We found that the DEGs consistently mapped to the phototransduction pathway (all comparisons, except seminal fluid vs. semen in experiment 1) and the neuroactive ligand-receptor interaction pathway (all semen and seminal fluid comparisons vs. controls; *Supplementary file 4*). Additionally, the Hippo signaling pathway was altered in both semen and seminal fluid comparisons against Hayes saline, the oxidative phosphorylation pathway only in semen vs. Hayes saline and seminal fluid vs. semen in experiment 2 (suggesting the production of ATP through this mitochondrial pathway is enhanced by the presence of sperm), the phagosome and tyrosine metabolism pathways exclusively in semen vs. Hayes saline, and the ribosome pathway in both seminal fluid vs. Hayes saline and seminal fluid vs. semen comparisons (with the pathway being consistently down-regulated in seminal fluid inseminated queens). The Hippo signaling pathway is known to control organ size during development by regulating cell-to-cell signaling and cell proliferation (*Halder and Johnson, 2011*), the phagosome pathway is linked to the process of particle engulfment by cells during inflammation, tissue remodelling, and defense against pathogens (*Stuart and Ezekowitz, 2005*), the tyrosine metabolism pathway can be involved in dopamine biosynthesis and plays a role in retinal pigmentation and associated diseases (*Molnár et al., 2005*; *Yang et al., 2017*), whereas the ribosome pathway is involved in protein synthesis, but also plays a role in DNA repair, replication, RNA processing and transcription, and development (*Yang et al., 2005*; *Lai and Xu, 2007*).

We consequently analysed in more depth the highly consistent effects we recovered across the two RNA-seq experiments for the phototransduction pathway - the process by which light is converted into electrical signals in photosensitive retinal cells (*Figure 2*). In *Drosophila* this process is mediated by a G-protein-coupled phospholipase C (PLC) that functions as the effector enzyme. This protein controls conductance changes in the plasma membrane of microvillar photoreceptors in the eye by activating two types of $Ca^{2+}$-permeable cation channels, TRP and TRPL (*Hardie, 2012*; *Figure 2*). We found that the gene coding for PLC, *no receptor potential A* (*norpA*), was always up-regulated in queens inseminated with semen or seminal fluid compared to control queens (*Figure 2* and *Figure 3*), which implies phenotypic effects on queen vision. We also found that semen and seminal fluid affected the expression of several genes whose *Drosophila* and honeybee orthologs are involved in the development of retinal microvilli (*Baumann and Lautenschläger, 1994*; *Hicks et al., 1996*), including an *Actin* gene (*Actin* a in *Figure 3*) and the *ninaC* gene. *Drosophila* null mutants at the *ninaC* locus have reduced amounts of visual pigment, defects in response termination and light adaptation, increased dark noise, and light-dependent retinal degeneration (*Hardie, 2012*). Finally, we found more variable expression changes for a series of other genes in the same pathway, including those coding for the cation channels TRP and TRPL and the production of diacylglycerol lipase (DAGL; *Figure 2*). This latter enzyme is known to catalyse the hydrolysis of diacylglycerol (DAG) into polyunsaturated fatty acids (PUFAs), which activates TRP and TRPL channels and is required for generating photoreceptor responses to light in *Drosophila* (*Hardie, 2012*).

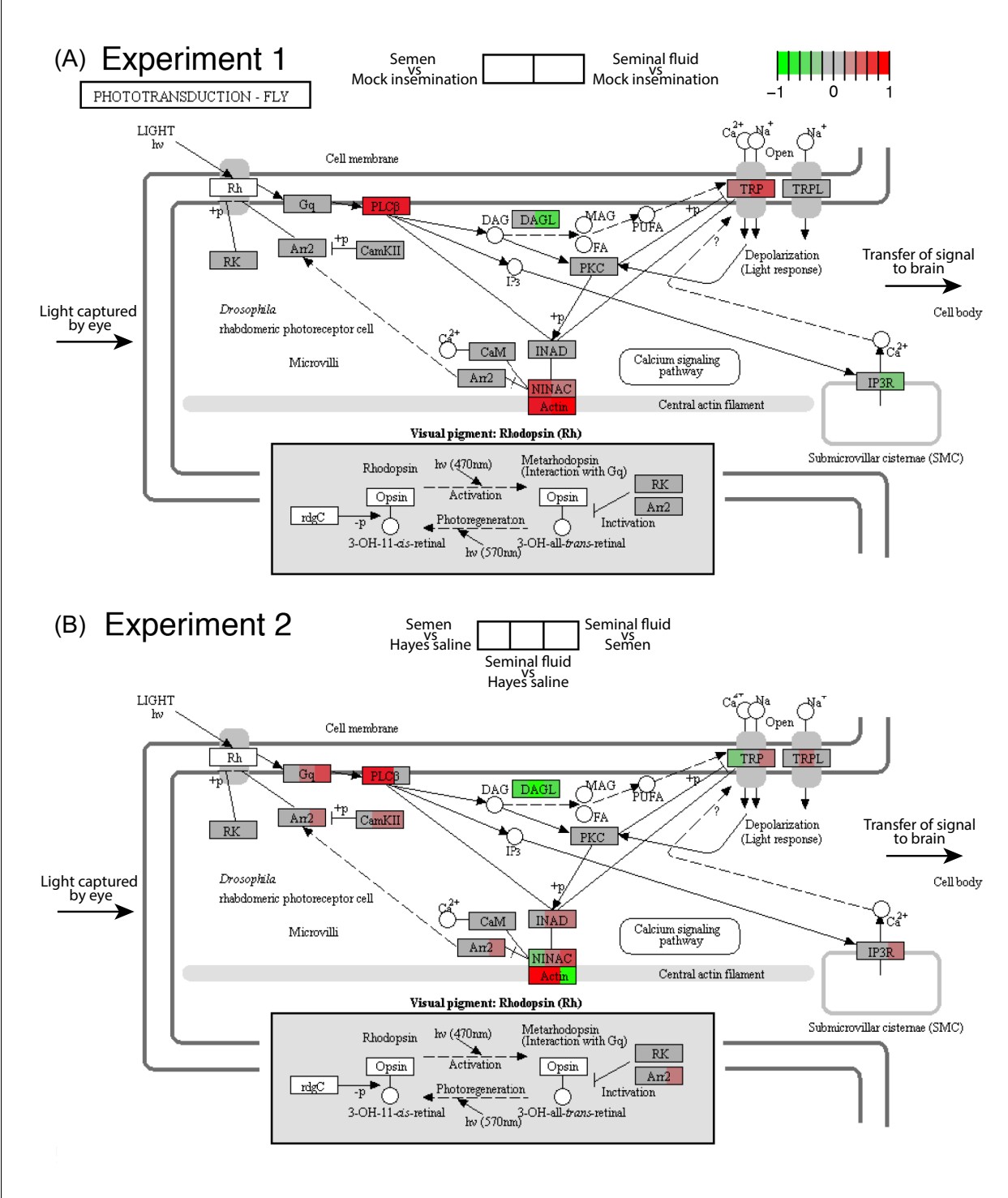

**Figure 2.** Seminal fluid and semen induce expression changes in honeybee queen brains for genes mapping to the phototransduction pathway of *Drosophila* in both Experiment 1 (**A**) and Experiment 2 (**B**). The plotted KEGG model represents the microvillus of a photoreceptor cell with the major proteins in the pathway shown by rectangles. Photoreceptors use the pigment rhodopsin (bottom grey panel) to absorb light, after which phospholipase protein C (PLC), upon activation via rhodopsin and the G-protein Gq, hydrolyzes phosphatidyl-inositol 4,5-bisphosphate (PIP2) to generate diacylglycerol (DAG), inositol 1,4,5-trisphosphate (InsP3) and a proton, resulting in the activation of two classes of $Ca^{2+}$-permeable cation channels, TRP and TRPL (***Hardie, 2012***) (top right). Several components of the cascade, including TRP, PLC, and protein kinase C (PKC), are assembled into multimolecular signaling complexes by the scaffolding protein INAD (***Hardie, 2012***), which has been suggested to be linked to the central F-actin filament via the ninaC class III myosin (***Hicks et al., 1996***) (bottom centre). The electrical impulses generated by light absorption reach the brain

*Figure 2 continued on next page*

*Figure 2 continued*

through the visual fibres of photoreceptor cells, which extend into the optic lobes (*Ehmer and Gronenberg, 2002*; *Wernet et al., 2015*). Results of Experiment 1 and Experiment 2 are shown in separate panels with quadrants within each protein-rectangle showing differences in the expression of the underlying coding genes of the first treatment group in each pair-wise comparison, relative to the second treatment group as shown above each of the figure panels (red for up-regulation, green for down-regulation, grey if no significant difference between treatments was detected but the gene was expressed in our datasets, white if the gene was not expressed).

DOI: https://doi.org/10.7554/eLife.45009.008

The following source data is available for figure 2:

**Source data 1.** Table of log2 (fold change) for each pair-wise comparison between treatment groups of genes mapping to the *Drosophila* phototransduction pathway.

DOI: https://doi.org/10.7554/eLife.45009.009

Across our two RNA-sequencing experiments, the annotation of the 37 genes with consistent differential expression in queens exposed to semen or pure seminal fluid inseminations compared to controls implies morphological changes in the retina (*Figure 1—figure supplement 2* and *Supplementary file 5*). This list included the genes *Hemicentin-1-like*, *Chaoptin*, *Thrombospondin*, and *Bardet-Biedl syndrome 2*, which have all been associated, among other roles, with retinal degenerative disorders in lineages ranging from *Drosophila* to humans (*Katsanis et al., 2000*; *Schultz et al., 2003*; *Stewart, 2006*; *Gurudev et al., 2014*). This effect may be significant because the gene *Hemicentin-1-like* consistently showed the greatest fold change between queens inseminated with semen or seminal fluid compared to the controls. Our functional enrichment analyses further highlighted several Biological Process terms suggesting changes in cell structure, cell adhesion and tissue morphogenesis, such as 'non-motile primary cilium assembly' (enriched in all our pair-wise comparisons), 'cell-cell junction assembly' and 'eye-antennal disc development', consistent with phenotypic effects on eyes or remodelling of brain structures (*Figure 1—figure supplement 3* and *Supplementary file 3*). The lists of significantly enriched Biological Process terms also contained genes related to the metabolism of cyclic guanosine monophosphate (cGMP; *Supplementary file 3*), a known derivate of phototransduction acting as regulator of ion channel conductance (*Hardie, 2012*). Finally, neuroactive receptor genes up-regulated in semen-treated and seminal-fluid-treated queens included the glutamate metabotropic receptor gene *mGlu$_2$R*, the glutamate receptor gene *NMDAR1*, and the serotonin receptor genes *5-HT$_2$alpha* and *5-HT1* (*Figure 3—figure supplement 1*, *Figure 3—figure supplement 2*, *Figure 3—figure supplement 3* and *Supplementary file 4*). However, *5-HT$_2$alpha* was only up-regulated compared to mock inseminations, suggesting this gene is under control of stretch receptors in the queen genital tract rather than affected by male-derived secretions.

To assess the extent to which our artificial insemination treatments produced changes similar to natural inseminations, we compared our DEGs with those of a previous study (*Manfredini et al., 2015*) that compared brain gene expression in naturally inseminated queens with that of virgin queens 48 hr after they were treated with $CO_2$ or not. Although we measured brain gene expression after 24 hr rather than 48 hr, 153 of the 1,327 DEGs were shared with the 1,050 DEGs identified in the natural insemination comparisons of *Manfredini et al. (2015)*. This 12–15% overlap (Hypergeometric test: representation factor = 1.7, p<0.0001; *Supplementary file 6*) was comparable to the degree of overlap in brain transcriptomes of newly inseminated honeybee queens across independent studies (*Kocher et al., 2010*; *Niño et al., 2011*; *Niño et al., 2013a*; *Manfredini et al., 2015*). We therefore concluded that this overlap in DEGs is biologically relevant and consistent with artificial insemination with seminal fluid or semen inducing part of the gene expression changes induced by natural insemination. Accepting this partial match as a replication of previous results is reasonable because (i) the difference in experimental design (i.e. different time-points and treatments) between our study and the previous one was substantial, (ii) the mating process includes factors other than seminal fluid known to affect queen brain transcriptomes (*Koeniger, 1976*; *Grozinger et al., 2007*; *Kocher et al., 2008*; *Kocher et al., 2009*; *Kocher et al., 2010*; *Niño et al., 2011*; *Vergoz et al., 2012*; *Niño et al., 2013a*; *Niño et al., 2013b*; *Manfredini et al., 2015*; *Brutscher et al., 2019*), and (iii) RNA-seq with limited replication is unlikely to identify all DEGs in a given tissue so the overlap would not be expected to be more than partial.

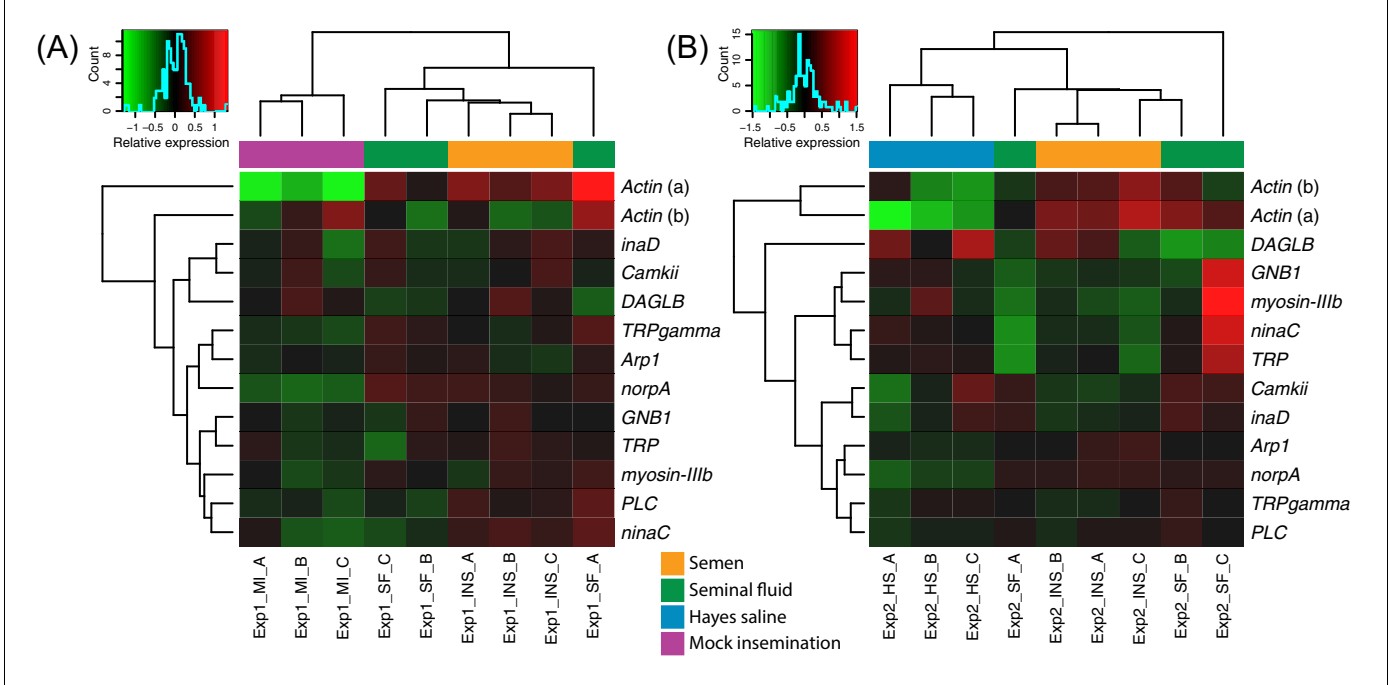

**Figure 3.** Heatmaps of gene expression for genes mapping to the phototransduction pathway in Experiment 1 (**A**) or Experiment 2 (**B**). Colours in each cell represent counts normalized by variance stabilizing transformation (*Huber et al., 2003*), which were then centred over the mean across samples. Only genes having expression differences between treatment groups above noise levels based on separate analyses performed with the essGene function (R Bioconductor package GAGE) in Experiment 1 or two are shown (legends to the right). Column and row dendrograms represent sample and gene clustering, respectively, based on Euclidean distances, reflecting that samples represent different treatments (central legend at the bottom) and that gene expression reacted differently to treatments. Small plots towards the top left of panels represent colour legends for expression values (x axis), and the distribution of gene counts (y axis) for these values across all genes and samples is depicted by blue lines. Bottom legends: MI = mock insemination, SF = seminal fluid insemination, INS = semen insemination, and HS = Hayes saline insemination.

DOI: https://doi.org/10.7554/eLife.45009.010

The following source data and figure supplements are available for figure 3:

**Source data 1.** Tables of normalized counts after variance stabilizing transformation and centring over the mean across samples for genes in the photo-transduction pathway showing expression changes over noise levels based on the essGene function in GAGE.

DOI: https://doi.org/10.7554/eLife.45009.014

**Figure supplement 1.** Differences in expression between pair-wise combinations of treatment groups of genes mapping to the KEGG neuroactive ligand-receptor interaction pathway for Experiments 1 and 2.

DOI: https://doi.org/10.7554/eLife.45009.011

**Figure supplement 2.** Heatmaps of gene expression for genes mapping to the KEGG neuroactive ligand-receptor interaction pathway in Experiments 1 (**A**) or 2 (**B**).

DOI: https://doi.org/10.7554/eLife.45009.012

**Figure supplement 3.** Receptor genes for, respectively, glutamate (first plot), serotonin (second and third plots), and N-Methyl-D-aspartic acid (last plot), showing consistent expression differences between insemination treatments in individual RNA-sequencing samples of honeybee brains in Experiment 1 (left panels) and Experiment 2 (right panels).

DOI: https://doi.org/10.7554/eLife.45009.013

The 153 shared DEGs with the study by *Manfredini et al. (2015)* were generally enriched for Biological Process terms related to energy metabolism (*Supplementary file 7*), but we assessed whether the overlapping genes included those related to vision more specifically. Out of the eight vision-associated genes listed in the Additional File 6 of *Manfredini et al. (2015)*, four (*ninaC, ninaA, norpA, chaoptin*) were also differentially expressed in our study, which represented a statistically significant overlap (Hypergeometric test: representation factor = 5.57, p=0.0002). A fifth gene (*Arr2*) was only altered in our seminal fluid vs. semen comparison in RNA-seq experiment 2 (*Figure 2*). Our systematic comparisons with the DEG lists of *Manfredini et al. (2015)* revealed another six joint DEGs with a potential role in vision (*Actin (a), Actin (b), GNB1, TRP, myosin-IIIb, Hemicentin-1-like*),

corroborating that the effects on vision between their natural inseminations and our artificial insemination experiments were similar. Taken together our gene expression results provide ample evidence that seminal fluid triggers the expression of vision-related genes similarly to what had been previously documented for naturally inseminated queens but without identifying the causal mating factors inducing these changes.

## Visual perception of queens after experimental exposure to seminal fluid

We obtained electroretinograms (ERGs) to explore the phenotypic eyesight consequences of gene expression changes in queen brains by investigating the temporal contrast sensitivity functions of queen eyes one and two days after artificial insemination with semen, seminal fluid or Hayes saline control solution. For the compound eyes, we found that queens inseminated with semen or seminal fluid showed lower response amplitude to flickering light of low temporal frequencies (the number of light flashes per second, expressed in Hz) than queens inseminated with Hayes saline, and that the magnitude of this effect increased on the second day after insemination (three-way statistical interaction term between stimulus frequency, number of days after insemination, and insemination treatment; N = 37, $\chi^2$ = 38.88, df = 8, p<0.0001; *Figure 4A* and *Supplementary file 8*; see *Figure 4—figure supplement 1* for the experimental set-up and *Figure 4—figure supplement 2* for an example of ERG response and details on how we derived contrast sensitivity measurements). To confirm that the increased reductions in compound eye performance over the two consecutive days were not due to queens having spent the night attached to their holders (see Methods for details), we compared the response amplitude 48 hr after insemination of the 18 queens that were measured both on day 1 and 2 with those of the nine queens that were only measured after 48 hr. We found no statistical difference (N = 27, $\chi^2$ = 8.566, df = 6, p=0.20; *Supplementary file 9*), suggesting the effects were exclusively due to seminal fluid exposure. The reduction in response amplitude remained statistically significant after excluding all measurements of the semen (full ejaculate) treatment (N = 25, $\chi^2$ = 15.16, df = 1, p<0.0001; *Supplementary file 10*), which is consistent with seminal fluid, rather than sperm, being primarily responsible for the reduction in phenotypic eyesight performance that we observed.

For the ocelli, we found that inseminations with semen or seminal fluid induced lower response amplitude than control treatments at higher light stimulus contrasts (statistical interaction between treatment and contrast; N = 35, $\chi^2$ = 16.53, df = 4, p=0.0024; *Figure 4B* and *Supplementary file 11*). Inseminating pure seminal fluid had the strongest effect, but treatment effects were less pronounced on the second day (interaction between treatment and day; N = 35, $\chi^2$ = 8.33, df = 2, p=0.0155; *Supplementary file 11*). When we repeated the statistical analysis after excluding all queens inseminated with semen, we still found a significant interaction between light stimulus contrast and insemination treatment, confirming that reduced ocelli visual perception is induced by seminal fluid and that this effect is most pronounced at higher light stimulus contrasts (N = 23, $\chi^2$ = 6.69, df = 2, p<0.0001; *Supplementary file 12*).

Low signal to noise levels precluded calculation of differences in light contrast sensitivity for the ocelli, but for the compound eyes the contrast sensitivity at low temporal frequencies was always highest when queens were inseminated with control saline solution and lowest when they were inseminated with semen (interaction between stimulus frequency and treatment; N = 37, $\chi^2$ = 24.05, df = 10, p=0.008; *Figure 4C* and *Supplementary file 13*). These effects appeared to develop over time, although the statistical interaction between day and treatment was only marginally significant (N = 37, $\chi^2$ = 11.02, df = 5, p=0.051; *Supplementary file 13*). However, after excluding all semen measurements there was no significant insemination treatment effect on contrast sensitivity, implying that the presence of sperm may be required to induce this effect (*Supplementary file 14*). We also measured the response of the compound eyes to isolated, short (1 ms) flashes of light, but this did not yield any significant differences in response duration, latency, or amplitude between treatment groups (*Figure 4—figure supplement 3*; see *Figure 4—figure supplement 4* for details on how amplitude, latency and duration were derived from the original ERG responses). Taken together our results offer compelling phenotypic evidence for the hypothesis that insemination induces reduced response to light stimulation in the compound eyes and, somewhat less consistently, in the ocelli of honeybee queens. As expected we found that most of these effects are primarily induced by seminal fluid and that sperm contributes to their enhancement.

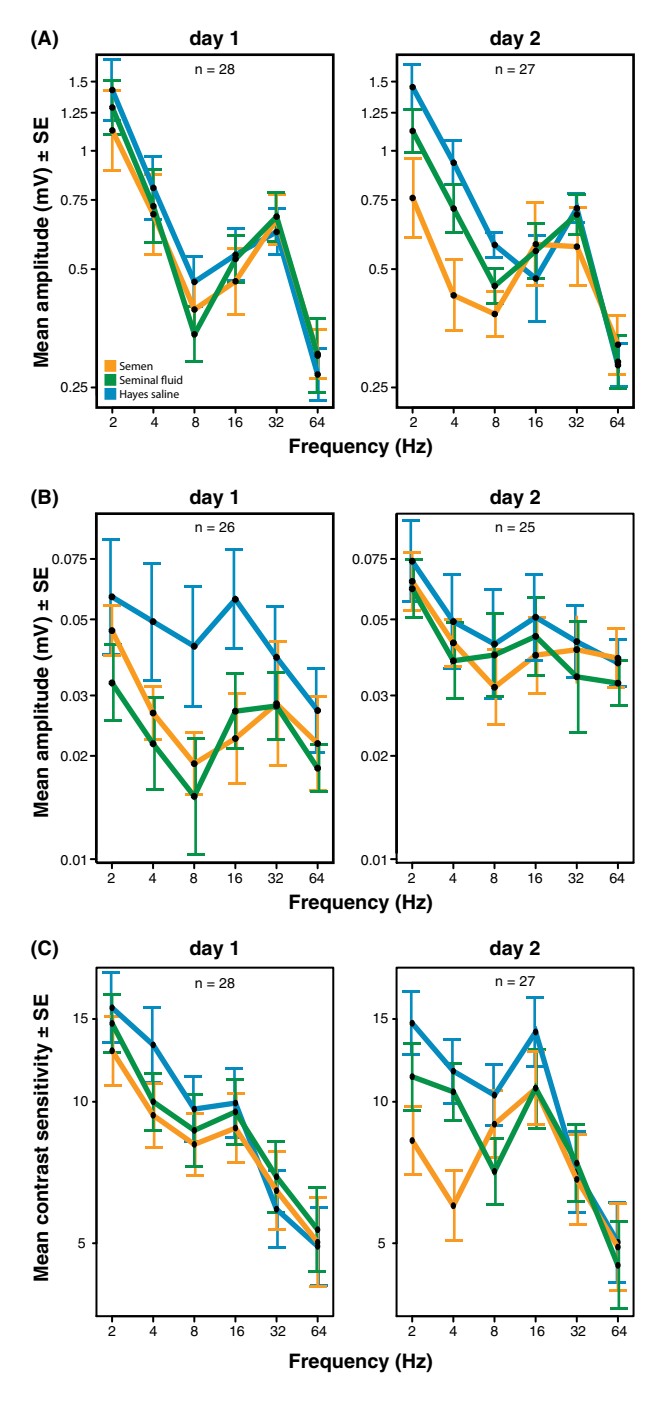

**Figure 4.** Response amplitude of electroretinograms (ERGs) measured from compound eyes (**A**) and ocelli (**B**) and mean contrast sensitivity (the lowest contrast to produce a detectable response) of ERGs from compound eyes (**C**) of honeybee queens stimulated with flickering lights of different frequencies (Hz). All measurements were taken one and two days after artificial inseminations with either semen, seminal fluid or Hayes saline control solution (see colour legend). Panel C includes all measurements taken at all stimulus intensities. Total sample sizes (n) are shown in the top centre of each panel and differ because some queens were exclusively measured at day 1 (N = 10; seminal fluid: N = 5, semen: N = 3, Hayes saline: N = 2) or at day 2 (N = 9; seminal fluid: N = 4, semen: N = 3, Hayes saline: N = 2), whereas the remaining queens were consecutively assessed on both days (N = 18; seminal fluid: N = 5, semen: N = 6, Hayes saline: N = 7). Two queens were excluded from the ocelli dataset because of large technical noise. See Materials and Methods for additional details.
DOI: https://doi.org/10.7554/eLife.45009.015

*Figure 4 continued on next page*

*Figure 4 continued*

The following source data and figure supplements are available for figure 4:

**Source data 1.** Electroretinography data recorded after stimulating queens' compound eyes and ocelli with flickering light.

DOI: https://doi.org/10.7554/eLife.45009.020

**Figure supplement 1.** Experimental set-up for electroretinography to measure honeybee queen eyesight responses to flickering light or short (1 ms) light impulses.

DOI: https://doi.org/10.7554/eLife.45009.016

**Figure supplement 2.** An example of response amplitude (mV) at different contrasts to calculate a contrast threshold measured at the highest intensity and temporal frequency of 4 Hz.

DOI: https://doi.org/10.7554/eLife.45009.017

**Figure supplement 3.** Response duration (A), latency (B) and amplitude (C) of electroretinograms (ERGs) measured from compound eyes of queens treated with semen, seminal fluid or Hayes saline after exposure to 1 ms light impulses.

DOI: https://doi.org/10.7554/eLife.45009.018

**Figure supplement 4.** Examples of impulse responses to 1 ms flashes of light in (A) median ocellus and (B) compound eye of honeybee queens.

DOI: https://doi.org/10.7554/eLife.45009.019

## Mating flight behaviour after artificial insemination with seminal fluid

We equipped 36 queens with radio-frequency identification (RFID) tags after artificial insemination with semen, seminal fluid or Hayes saline and monitored their natural flight activity over several consecutive days. Among the 34 queens that left their hives, those inseminated with either pure seminal fluid or semen were more likely to get lost, i.e. they did not return to their hives when they flew again. They also triggered hive entrance sensors more often than their sister queens inseminated with Hayes saline (*Figure 5A*; Binary Logistic Regression of "Not Found' by "Treatment' and "PingNr', the acronym for the number of times queens triggered the sensors located at hive entrances;

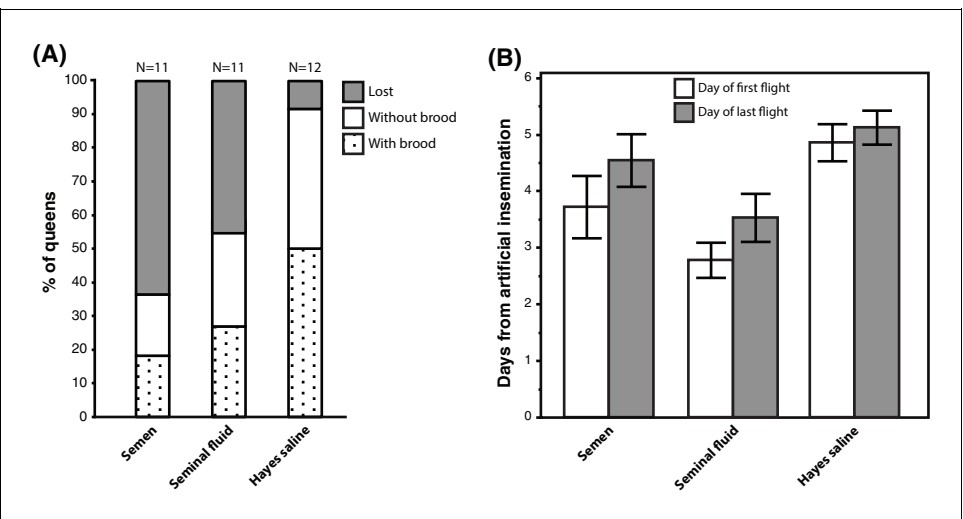

**Figure 5.** Seminal fluid and semen induce alterations of mating flight behaviour in honeybee queens. (A) Stacked bars showing the percentage of queens in each artificial insemination treatment that were lost and how many of them produced brood after returning from their last mating flight, with absolute sample sizes printed on top of the bars. (B) Effects of insemination and control treatments on the time from artificial insemination to making their first and last flight. Bars show mean ±SE.

DOI: https://doi.org/10.7554/eLife.45009.021

The following source data is available for figure 5:

**Source data 1.** Experimental data recorded during the mating flight experiment.

DOI: https://doi.org/10.7554/eLife.45009.022

Nagelkerke $R^2$ = 0.535, 79.4% Correct; Treatment: N = 34, Wald-$\chi^2$ = 6.970, df = 2, p=0.031; PingNr: N = 34, Wald-$\chi^2$ = 4.843, df = 1, p=0.028). The latter effect is consistent with semen- and seminal-fluid-inseminated queens being disoriented or distressed by sunlight and spending more time at hive entrances than control queens. However, this difference may also reflect increased general activity induced by seminal fluid, similar to the sex peptide increasing female activity and reducing siesta sleep in *Drosophila* (*Isaac et al., 2010*). Of a total of 13 queens that did not return to their hive, four triggered sensors for the last time two days after insemination (seminal fluid: N = 2, semen: N = 2), two after three days (seminal fluid: N = 1, Hayes saline: N = 1), four after four days (seminal fluid: N = 1, semen: N = 3), and three between day six and seven (seminal fluid: N = 1, semen: N = 2).

Of the 21 queens returning to their hives, 17 (81 %) performed flights that exceeded 7 min, which is a conservative threshold for assessing a complete mating flight based on the shortest mating flight performed by Hayes-inseminated control queens that later laid eggs in our study (see Materials and Methods for details). We confirmed the presence of brood in worker comb in 11 of these queens (52%), indicating that they originated from fertilised eggs, because unfertilised eggs are laid into larger drone combs. Of these 17 queens attempting real mating flights, those inseminated with seminal fluid or semen left their colonies 1–2 days earlier than those inseminated with Hayes saline (ANOVA, N = 17, df = 2, p=0.026). This effect appeared to be general because we found the same statistical result when we analysed data of all 34 queens: those inseminated with either seminal fluid or semen triggered sensors for the first time earlier than control queens (*Figure 5B*; Kruskal-Wallis, N = 34, df = 2, p=0.004). Finally, semen- or seminal-fluid-inseminated queens also triggered sensors for the last time earlier than saline-treated queens (*Figure 5B*; Kruskal-Wallis, N = 34, df = 2, p=0.033).

## Discussion

We conducted a series of genetic and phenotypic experiments and found that seminal fluid induces substantial gene expression changes in the brains of honeybee queens and reduces their visual performance within 24–48 hr. Consistent with these effects, a controlled apiary experiment with treatment and control sister queens showed that queens receiving seminal fluid (either pure or as part of semen) were more likely to get lost during mating flights in spite of embarking on their flights earlier than control queens, as would be expected from queens actively responding to a perceived deterioration of their visual sensitivity.

We confirm findings of previous studies that documented comparable expression changes of genes involved in visual perception in honeybee queen brains after natural inseminations (*Kocher et al., 2008*; *Kocher et al., 2010*; *Manfredini et al., 2015*). Our results are also comparable to earlier findings in *Drosophila,* reporting differential gene expression effects on brain phototransduction pathway induced by insemination (*Gioti et al., 2012*). These effects are triggered by the sex peptide (*Gioti et al., 2012*), although the phenotypic consequences for female vision were not investigated in fruit flies. Similar post-mating changes in the expression of phototransduction genes were also found in females of the egg parasitoid wasp *Anastatus disparis* (*Liu and Hao, 2019*) and in *Bombus terrestris* bumblebee queens (*Manfredini et al., 2017*). In all these species males appear to manipulate the copulation behaviour of females by reducing their re-mating rates via manipulative effects of ejaculate compounds (*Chen et al., 1988*; *Baer et al., 2001*; *Liu and Kubli, 2003*; *Chapman et al., 2003b*; *Liu and Hao, 2019*). In *B. terrestris*, where the seminal fluid component linoleic acid significantly reduces a queen's willingness to re-mate (*Baer et al., 2001*), gene expression changes induced by insemination not only affected the phototransduction pathway but also involved neuroactive ligand-receptor interactions, the Hippo signaling pathway and the phagosome pathway (*Manfredini et al., 2017*). We found in our study that all of these pathways were also altered in the brain of honeybee queens after seminal fluid exposure. This suggests that the mechanisms that we identified may be conserved across bee lineages and possibly across the Hymenoptera or the insects in general, and that males appear to use seminal fluid components to manipulate female mating frequencies to promote their own fitness interests. We therefore interpret the rapid loss of visual perception and the clear survival costs for artificially inseminated queens during natural mating flights as a manipulative strategy of already stored ejaculates to reduce queen tendencies to engage in additional mating flights. In the paragraphs below, we evaluate the likelihood of

alternative explanations for our findings and point out opportunities for future hypotheses-driven research aiming to clarify the causal mechanisms that mediate such sexual conflicts.

## Reductions of female visual perception after mating

One could argue that the loss of queen visual perception after insemination allows queens to reduce the substantial physiological costs for the maintenance of a metabolically complex trait (*Niven and Laughlin, 2008*) and that the effects we documented would be adaptive if there are trade-offs between energetically demanding life history traits. Honeybee queens return to a hive where levels of luminance are low, so they do not require fully functional eyesight because all communication becomes non-visual and mediated by pheromones, direct contact, vibrations or sounds (*Billen, 2006*). Energetic trade-offs might be specifically important for newly inseminated honeybee queens, because their ability to produce large numbers of fertilised eggs shortly after their mating flight(s) is crucial to compensate for the lack of egg production in the hive and for the workers that were lost when the old mother queen left with a swarm a few days before (*Winston, 1987*; *Grozinger et al., 2014*). Because older honeybee queens still have the capacity to initiate a new colony through swarming (*Winston, 1987*), a permanent loss of eyesight is not expected, even though swarming flights are probably less visually demanding than mating flights because swarming queens are always accompanied and guided by a large number of workers.

If queens have to trade-off energetically demanding physiological traits such as visual perception and fecundity, they should be able to determine the optimal timing to modify the amount of energy allocated to these traits through mechanisms that provide reliable cues about the quantity, quality and diversity of the sperm they obtained, such as monitoring the number of copulations achieved, the degree of filling of the lateral oviducts with semen and the extent to which sperm have entered the spermatheca. However, our results show that the alterations in queen visual perception even occur in queens that only receive seminal fluid, which by itself does not deliver such information to queens. This is consistent with a conflict-mediating role of seminal fluid and not with queens harmoniously managing their optimal number of inseminations. We therefore conclude that the most parsimonious explanation of our findings is that seminal fluid inhibits future promiscuity of queens as part of a sexual conflict, because: 1. We could show that changes in visual perception start much earlier than would be predicted if these changes were merely adaptive to queens, i.e. during the time window of repeated nuptial flights (*Winston, 1987*), and that visual perception is reduced well before the ca. 40 hr that sperm cells need to reach final storage in a queen's spermatheca (*Woyke, 1983*). 2. Our results confirmed that seminal fluid without sperm is capable of triggering both the genetic and the behavioural changes in the brains of honeybee queens, similar to those reported after natural mating flights (*Grozinger et al., 2007*; *Kocher et al., 2008*; *Kocher et al., 2010*; *Manfredini et al., 2015*). 3. We also confirmed that these seminal fluid effects impose survival costs on queens during a period in which they would be expected to possess an unimpaired ability to engage in additional mating flights and return to the hive with high probability.

## Sexual conflict over the number of mating flights

A key condition for the presence of a sexual conflict and an arms race between queens and drones over paternity is that both sexes should pay fitness costs. Our results suggest this to be the case because successful ejaculates from a first mating flight (in pre-storage for up till 40 hr) risk collective death when their seminal fluids handicap queens in attempting additional flights. Queens have been under selection to accept mortality risks emanating from additional flights because of well-documented fitness benefits of permanently storing sperm of sufficient genetic diversity (*Mattila and Seeley, 2007*; *Mattila et al., 2012*). However, their pre-stored ejaculates, competing within the queen sexual tract to obtain permanent storage in the spermatheca, are selected to take higher risks of queen-failure because their future paternal fitness will be reduced in a way that is proportional to the amount of sperm their mate obtains in a subsequent mating flight, especially when queens only store 3–5% of the sperm they received (*Baer, 2005*). Average mating frequencies of honeybee queens can thus be expected to be a compromise between the need for queens to obtain additional genetic diversity for their colonies and the mortality costs associated with additional flights resulting from longer exposure to predators, an increased risk of getting infected with sexually transmitted parasites (*Peng et al., 2016*) and aggravated by male manipulation of their visual perception.

Where the arms race equilibrium settles will quantitatively depend on a number of environmental and genetic factors that may vary between geographic regions, seasons and species or subspecies of honeybees (*Kraus et al., 2004*; *Kraus et al., 2005*; *El-Niweiri and Moritz, 2011*). The sexual conflict hypothesis assumes that queens will have evolved mechanisms to neutralise seminal fluid compounds that affect their visual perception and that they vary in their effectiveness of counter-mechanisms to neutralise drone manipulations. The behaviours that we recorded in our apiary experiment and the fact that we detected substantial variation in queen visual perception one day after artificial inseminations provide further support for our interpretation of a sexually antagonistic arms race between drones and queens. These interactions would thus resemble the rapid evolution of adaptations and counter-adaptations in reproductive fluid molecules in other organisms (*Chapman, 2001*; *Swanson and Vacquier, 2002*; *Andrés et al., 2006*; *Panhuis et al., 2006*; *Haerty et al., 2007*; *Findlay et al., 2009*; *Walters and Harrison, 2010*). The interpretation of our results as being consistent with an ongoing sexual arms race over the number of mating flights rather than the number of copulations per se also agrees with previous research suggesting that honeybee queens adjust their flight number based on their insemination success during (a) previous flight(s) (*Schlüns et al., 2005*). These studies already suggested that natural selection should act primarily at the level of queen flights, which represent greater efforts and risks than individual copulations occurring in quick succession do (*Tarpy and Page, 2000*; *Schlüns et al., 2005*). The physiological and/or mechanical mechanisms mediating these responses remain poorly understood and the conceptual logic of our present study provides a novel framework and clear incentive for unravelling them.

## Further considerations, caveats and suggestions for future research

Although our genetic, phenotypic and behavioural results are all consistent with changes in queen vision, more research is needed to confirm whether the reduced performance of queens during mating flights is a consequence of perturbed visual perception only, or whether other physiological effects induced by seminal fluid might also play a role. One could argue that reduced queen survival in our apiary experiment resulted from harmful effects of seminal fluid on the general health of queens. This could for example occur if seminal fluid components affected female immunity as a form of 'collateral damage' of enhancing egg production (*Rolff and Siva-Jothy, 2002*; *Short et al., 2012*). However, such arguments typically refer to studies of seminal fluid effects in non-social insects, where females re-mate at regular intervals throughout adult life. In these species, traits that enhance the paternity share of a focal male in the present clutch at the expense of later female health can evolve, because these female's survival costs do not diminish the focal male's fitness return (*Chapman et al., 1995*; *Johnstone and Keller, 2000*; *Rolff and Siva-Jothy, 2002*; *Kemp and Rutowski, 2004*; *Wigby and Chapman, 2005*). However, a series of review papers and experimental studies have shown that this type of health effects are neither expected nor found in social insect queens (*Tsuji et al., 1996*; *Boomsma et al., 2005*; *Schrempf et al., 2005*; *Heinze and Schrempf, 2008*; *Lopez-Vaamonde et al., 2009*; *Rueppell et al., 2015*; *Barribeau and Schmid-Hempel, 2017*). Natural selection will consistently eliminate traits mediating such 'collateral damage' because queens need to produce many cohorts of sterile workers before their colony can produce winged reproductives. This implies that even a slight negative effect of mating on the general physiological performance of queens would likely preclude survival until first reproduction and make males lose their paternity success together with the queen they inseminated. Hence, both the presence of seminal fluid effects on general health in female fruit flies and the absence of such effects in queens of social insects are direct consequences of the different evolutionary origins and temporal distribution of life-time female promiscuity. The exceptional mating system of the honeybee allows sexual conflict to be expressed for just a few days and can only target the odds of successful additional female promiscuity during this narrow time window, not their subsequent state of health.

Although the ultimate (evolutionary) logic of an arms race over queen promiscuity during additional mating flights seems compelling and consistent with the evidence so far, it is also important to evaluate the degree to which our study provides insights into the proximate causation factors involved. Our data provide first and solid proof of concept evidence, but further work will be needed to unravel the complex interactions between differential gene expression and phenotypic effects on visual perception and flight behaviour. Our electrophysiological experiments for the compound eyes were consistent with visual perception loss accumulating quickly and gradually, and our apiary experiment showing that queens embarked on additional flights earlier, as if they actively responded to

the fact of becoming visually handicapped, also matched our expectations. However, this pattern was different for the ocelli, suggesting not all complexities of seminal fluid effects on queen mating behaviour are straightforward. The drone congregation areas that honeybee queens need to localise in flight (*Winston, 1987*) typically occur in spatially restricted areas, often associated with specific land-marks (*Galindo-Cardona et al., 2012*) and have a diameter of 30–200 m (*Baudry et al., 1998*). Well-functioning compound eyes and ocelli thus appear to be essential for reaching drone congregations and for returning back to the hive. Honeybee workers use path integration with reference to the sun and a mental map based on learned visual landmarks when they navigate away from and back to the hive (*Menzel et al., 2000*; *Menzel et al., 2005*), but further research to confirm similar abilities in queens is required, if only because virgin queens have no previous flight experience to draw upon.

Further research should also quantify whether there is a trade-off between cumulative mating flight effort and the initiation and scale of early egg-laying, because both are likely to affect the future reproductive success of queens. Additional causal factors that determine mating flight decisions of queens may emerge from such work, because our differential gene-expression analyses identified, apart from changes in phototransduction pathway genes, also a number of genes involved in energy metabolism, regulation of phagocytosis, DNA replication, RNA transcription, protein synthesis, and cellular adhesion, suggesting that metabolic effects and structural modifications or deterioration in photoreceptors or neurons may also occur. We also note that, although the queens used for our apiary experiment received a total volume of seminal fluid comparable to what they would likely receive during natural mating flights, the genetic diversity of these seminal fluid mixtures was likely higher compared to natural inseminations because we used hundreds of drones to collect the pure seminal fluid samples. We therefore generated qualitative evidence in the direction of our predictions, but we do not know whether the quantitative outcomes of our experiments reflected natural circumstances. For example our apiary experiments may have produced higher queen mortality than natural mating because queens were unusually handicapped by a larger diversity of seminal fluid molecules. Replication of our manipulative apiary experiment will therefore be needed to obtain a better quantitative understanding of the arms race dynamics for which we provide the first evidence. Finally, although we were careful to only expose queens to minimal amounts of $CO_2$ and we were successful in recovering clear differences in gene expression between our treatment and control groups, the use of $CO_2$ to stimulate ovary activation and to narcotise queens during artificial insemination may have masked additional changes induced by seminal fluid if they are also induced by $CO_2$ exposure. Future research should clarify whether such confounding effects exist. Overall, our findings underline that polyandrous social insects provide intriguing testbeds for general sexual conflict theory and that honeybees offer a plethora of interesting research opportunities to unravel the proximate mechanisms that shape the practical implementation of the sexual conflict that we documented.

## Materials and methods

### Queen rearing

All queens used for experiments were bred at the University of Western Australia according to standard apicultural practices (*Laidlaw and Page, 1997*). We grafted honeybee (*Apis mellifera ligustica*) larvae at day four of their development (i.e. one day after hatching) from a single colony, transferred them into plastic queen cells (Ceracell, Aukland, NZ) and placed them into a queen-right cell-building colony, prepared 24 hr in advance by moving two frames of uncapped brood above the queen excluder and placing the graft bar in between these brood frames. Ten days later we placed the developing queens in their cells into 4-frame queen-less nucleus hives with queen excluders at the entrance to prevent natural mating flights and allowed them to hatch. Four days after hatching and one day before artificial inseminations, virgin queens were removed from their hives, caged, and exposed to $CO_2$ for 1 min to stimulate ovary activation. Five nurse bees were added to the cages before returning queens to their original nucleus hives. Queens used in the apiary mating flight experiment were never exposed to $CO_2$ but were cooled on ice prior to, and during, artificial inseminations (see below).

## Collection of semen and seminal fluid for artificial insemination

We collected semen (consisting of seminal fluid and spermatozoa) or pure seminal fluid using a pre-viously-developed protocol (*Baer et al., 2009*) and keeping the collection procedures identical across all experiments performed. We randomly caught drones at hive entrances and kept them in cages before transferring them in two foster hives. After a maximum of two days, we re-collected the cages and anaesthetized drones with chloroform to initiate ejaculation. We then squeezed the drone's abdomens between two fingers and collected the semen appearing on the tip of the endo-phallus in a glass capillary connected to a syringe (Schley, Germany) (*Baer et al., 2009*) and immedi-ately used it for artificial inseminations for the 'semen' treatments. To obtain pure seminal fluid, we collected semen in batches of several hundred drones (~2000 in total) as described above and pooled these samples in 1.5 ml Eppendorf tubes, which we centrifuged at 18,500 g and 4℃ for 25 min. The supernatants were collected into new 1.5 ml Eppendorf tubes and centrifuged for 10 min at 18,500 g and 4℃ (*Baer et al., 2009*). We then collected the seminal fluid as supernatant, froze these aliquots in liquid nitrogen and stored them at −80℃ prior to further experiments. This centri-fugation method has previously been shown to be a reliable method to collect seminal fluid uncon-taminated by sperm cells and/or major sperm proteins (*Baer et al., 2009*).

## Artificial insemination procedure for RNA-sequencing

Artificial inseminations were performed as described in detail earlier (*Mackensen and Roberts, 1948*). For a first RNA-sequencing experiment we compared brain gene expression in queens that we artificially inseminated with either semen, seminal fluid or a mock insemination treatment, where no fluid was injected into the vaginal orifice. Virgin queens were sampled from their hives and ran-domly assigned to one of three experimental groups (referred to as 'Experiment 1'): (i) instrumen-tally inseminated with 6 µl of semen pooled from approximately 10 males, (ii) instrumentally inseminated with 6 µl of seminal fluid, and (iii) mock-inseminated without injecting any fluid. In a sec-ond RNA-sequencing experiment (referred to as 'Experiment 2') we further controlled for insemina-tion of liquid into the queen reproductive tract and assessed to what extent gene expression changes in queen brains were dependent on reception of semen or seminal fluid rather than on the mechanical stimulation of the reproductive tract upon insemination. To do this, we randomly assigned queens to one of three experimental groups: (i) instrumentally inseminated with 6 µl of semen, (ii) instrumentally inseminated with 6 µl of seminal fluid, and (iii) instrumentally inseminated with 6 µl of Hayes saline (9 g NaCl, 0.2 g $CaCl_2$, 0.2 g KCl and 0.1 g $NaHCO_3$ in 1000 ml $H_2O$, adjusted to pH 8.7 and sterilised by filtration through a 0.22 µm syringe-filter, Membrane Solutions).

To artificially inseminate queens, we sedated them with $CO_2$ for a few seconds before placing them in a holder mounted onto a standard artificial insemination instrument (Schley, Germany) and inseminating them according to the treatments. Queens were afterwards allowed to recover for about 30 min before we returned them to their hives. We recollected queens after 24 hr, narcotised them with $CO_2$ for a few seconds and flash froze their heads in liquid nitrogen. All heads were stored at −80℃ until dissections.

## Brain collection and RNA extraction

Brains of queens were dissected with Inox five watchmaker forceps under an Olympus SZX10 stereo microscope in ice-cold sterile Hayes saline. We ensured that brains were removed intact, including the optic lobes, and that all hypopharyngeal and other extraneous glandular tissue was removed. We combined the brains from three individual queens to obtain enough RNA for Illumina TruSeq sequencing (see below), froze these pooled brain samples in liquid nitrogen and immediately stored them at −80℃. We consequently obtained three biological replicates (each consisting of three pooled brains) per treatment group in both experiments 1 and 2 (18 samples in total; *Supplementary file 15*). To extract RNA from pooled brain samples, we briefly thawed them on ice, placed them in 10 µl of 0.25 M Tris pH 7.5 and homogenised them with a plastic pestle. We then added 50 µl of Trizol to each sample, incubated samples on ice for 15 min, thoroughly vortexed and returned them to ice for another 15 min, followed by centrifugation at 20,000 g for 15 min at 4℃. In the next step, we collected the supernatant, added one volume isopropanol, briefly vortexed the samples and incubated them at room temperature for 20 min, followed by centrifugation at 20,000 g for 15 mins at 4℃. After discarding the supernatants, we washed each pellet with 70% EtOH,

followed by air-drying and resuspension in 20 µl of RNase-free $H_2O$. To precipitate the RNA, we added 1.6 volumes of ice-cold 4 M LiCl and incubated samples on ice for 1 hr, followed by centrifugation at 20,000 g for 15 min at 4°C. After discarding the supernatants, each RNA pellet was washed with 500 µl 70% EtOH before resuspension in 10 µl RNase-free $H_2O$.

## Library preparation and RNA-sequencing

Sequencing libraries were generated from 1 µg total input RNA using the TruSeq Stranded mRNA Sample Kit (Illumina, San Diego, CA) and single-end sequencing by synthesis was performed on a HiSeq 2500 (Illumina, San Diego, CA) for 120 cycles, thus generating 120 bp reads. Samples were dispersed over four lanes of a single plate. The sequencing produced a mean of 54,574,574 reads per sample (range 38,806,853–74,375,492; *Supplementary file 15*). RNA-sequencing data have been deposited in NCBI's Gene Expression Omnibus (*Edgar et al., 2002*) and are accessible through GEO Series accession number GSE127185 (https://www.ncbi.nlm.nih.gov/geo/query/acc.cgi?acc=GSE127185).

## Gene expression analyses

Reads were quality-controlled using FastQC (http://www.bioinformatics.babraham.ac.uk/projects/fastqc/) and subsequently processed with Trimmomatic (*Bolger et al., 2014*) to remove adapters and low quality bases using the following parameters: LEADING: 3 (trim the leading nucleotides until quality >3), TRAILING: 3 (trim the trailing nucleotides until quality >3), SLIDINGWINDOW: 4:15 (trim the window of size four for reads with local quality below a score of 15), and MINLEN: 36 (discard reads shorter than 36 bases). On average, 1.7% of the total reads were discarded during this step. Next, we removed reads that matched ribosomal RNA sequences (rRNA) using SortMeRNA (*Kopylova et al., 2012*), which implied discarding an average of 2.3% of the reads. The remaining reads were aligned with STAR v. 2.4.2a (*Dobin et al., 2013*) to the latest version of the honeybee genome (*Apis mellifera* assembly 4.5) available on BeeBase (http://hymenopteragenome.org/beebase/?q=download_sequences), which resulted in an average mapping rate of 97% (*Supplementary file 15*). Mapped reads were converted into raw read counts with the htseq-count script (http://www.huber.embl.de/users/anders/HTSeq/doc/count.html), and the R (*R Core Development Team, 2015*) Bioconductor (*Huber et al., 2015*) package DESeq2 v.1.10.1 (*Love et al., 2014*) was subsequently used to quantify differential gene expression between all pair-wise combinations of treatment groups for each experiment. *P* values of differential expression analyses were corrected for multiple testing with a false discover rate (FDR) of 10%.

We used Non-metric Multidimensional Scaling (NMDS) of Bray-Curtis dissimilarities in Paleontological Statistics v.3.04 (*Hammer et al., 2004*) to investigate overall sample clustering after removing experimental batch effects with the removeBatchEffect function implemented in the R Bioconductor package edgeR v.3.12.1 (*Robinson et al., 2010*). A Hypergeometric test (http://nemates.org/MA/progs/overlap_stats.html) was used to quantify overlap in differentially expressed genes between our study and *Manfredini et al. (2015)*. Conducting such a comparison was of interest to assess whether our artificial insemination treatments induced similar effects as those found in naturally inseminated queens. *Manfredini et al. (2015)* performed their experiments on Australian honeybees, and quantified gene expression in the brains of (i) virgin queens, (ii) naturally-inseminated queens, and (iii) $CO_2$-treated queens two days after treatments were performed. They also used RNA-sequencing rather than microarrays as in previous studies (*Grozinger et al., 2007*; *Kocher et al., 2008*; *Kocher et al., 2010*; *Niño et al., 2011*; *Niño et al., 2013a*). For all of the above reasons the Manfredini data sets were the most appropriate to compare our results with those obtained from natural mating comparisons.

## Gene Ontology (GO) and pathway analyses

To perform Gene Ontology (GO) enrichment analyses we first annotated the honeybee transcriptome (OGS v3.2 available on BeeBase) by running BLASTx against the NCBI Non-Redundant (NR) database (performed in March 2016), retaining the first 20 hits with a cutoff eValue of $10^{-3}$. Blast2GO v.3.2 (*Conesa et al., 2005*) was used to map the ensuing annotations to GO terms. We then used a hypergeometric test implemented in the R Bioconductor package GOstats v.2.36.0 (*Falcon and Gentleman, 2007*) to evaluate the differentially expressed gene lists for GO term

associations, using the full transcriptome as background and retaining Biological Process and Molecular Function terms with *P* values < 0.05. REVIGO (*Supek et al., 2011*) was subsequently used to reduce redundancy in significant GO terms and to summarise results by semantic similarity. Perturbed genetic pathways were identified with the R Bioconductor package Generally Applicable Gene-set Enrichment for Pathway Analysis (GAGE v.2.20.1) (*Luo et al., 2009*) by retaining Kyoto Encyclopedia of Genes and Genomes (KEGG) signaling and metabolic pathways (accessed in August 2016) with *q* values < 0.2. For significant pathways, we identified genes that showed expression changes over noise levels with the essGene function in GAGE using default parameters.

## Electroretinogram (ERG) measurements of queen visual perception

To test whether exposure to seminal fluid resulted in a phenotypic alteration of queen visual perception, we reared virgin queens as described above and artificially inseminated them with either: (i) 6 µl of semen, (ii) 6 µl of seminal fluid or (iii) 6 µl of Hayes saline (see above for details). After the insemination procedure, we caged queens individually and randomly placed them back into one of two foster colonies. The following day we recollected the queens and sedated them on ice, removed their legs and fixed them with bee wax on a plastic holder to minimise head movements. The holders with the queens were randomly assigned to, and mounted in, one of two Faraday cages.

We recorded ERGs from both the queens' compound eyes and their median ocellus using a differential amplifier (DAM50, World Precision Instruments) connected to a standard PC via a 16-bit data acquisition card (USB-6353, National Instruments). All recordings were controlled by custommade software in MATLAB R2014a (*Source code 1*; *Ogawa et al., 2015*). A silver/silver-chloride wire of 0.1 mm diameter was inserted into the animal's thorax and served as the reference electrode. The recording electrode was a platinum wire of 0.254 mm diameter covered with conductive, neutral pH gel (ECGEL250, Livingstone International), carefully positioned on the dorsal surface of one of the compound eyes or along the median ocellar lens (*Figure 4—figure supplement 1*). The electrical ground was connected to the Faraday cage. The light source was a 'cool white' LED light with 5 mm diameter (C503C-WAS-CBADA151, Cree Inc, Durham, NC, USA), powered by a custommade LED driver using pulse width modulation (PWM). All light stimuli were checked for linearity using a calibrated light metre (ILT1700, International Light Technology). The LED was positioned at an elevation of approximately 30° in the queen's visual field and kept at a constant distance of 70 mm from the queen's head. To reduce any electrical noise from the light source, two grounded metal shields with 3 and 1 mm holes were positioned 30 mm from the light source and 10 mm from the queen's head, respectively. A total of 37 queens were tested, 12 inseminated with semen, 14 with seminal fluid and 11 with Hayes saline. From a total of 28 queens that were used to measure visual perception one day after the insemination treatments, 18 were re-used for measurements a day later – that is two days after they were artificially inseminated, and kept attached to their holders in a small plastic container in the dark overnight after pipette-feeding them with sugar water. Another nine queens were only measured two days after insemination and were collected directly from the hives where they had been placed after the inseminations.

Queens were dark-adapted for 20 min prior to all recordings. To measure the eyes' ability to detect temporal changes in brightness, we measured the temporal contrast sensitivity function, which is the inverse of the lowest detectable contrast at each temporal frequency. The stimulus contrasts were expressed as Michelson contrasts $\frac{L_{MAX}-L_{MIN}}{L_{MAX}+L_{MIN}}$ where $L_{MAX}$ is maximum light intensity and $L_{MIN}$ is minimum light intensity of the square wave stimulation pattern. We used three light intensities ($2.74*10^{-2}$ W/cm$^2$, $2.74*10^{-3}$ W/cm$^2$, $2.74*10^{-4}$ W/cm$^2$; we also used a second Faraday cage/ light source with 70% dimmer LED intensities, and randomly assigned queens to these two set-ups), and we tested all 80 combinations of eight temporal frequencies (2, 4, 8, 16, 32, 64, 128, 256 Hz) and 10 contrasts (0.0019, 0.0039, 0.0078, 0.0156, 0.0312, 0.0625, 0.125, 0.25, 0.5, 1) at each light intensity. For an example of ERG response and further details on how we derived contrast sensitivity measurements see *Figure 4—figure supplement 2*. We next recorded the impulse response of the compound eyes and ocelli to a 1 ms flash of light, at the same three light intensities as before, followed by 2 s of darkness. An averaged response of 100 times repetitions was taken as the impulse response for each individual. The average response per condition was then analysed for its latency, duration, and amplitude (see *Figure 4—figure supplement 4* for an example of original ERG

response and further details on how amplitude, latency and duration were derived from the original responses).

## Statistical analyses for electroretinogram (ERG) measurements

To test for significant differences between treatments in ERG measurements, we used linear mixed effects models within the R package lme4 (*Bates et al., 2015*). All models included animal identity, date of measurement, and recording Faraday cage as random effects to account for repeated measures of some queens, for measurements performed on different days, and for measurements having been recorded in two different Faraday cages. The dependent variable contrast sensitivity was analysed as a function of the fixed effects: number of days after insemination, stimulus intensity, temporal frequency, contrast, and treatment group. The dependent variables amplitude, latency, and duration of the impulse response to a brief 1 ms light pulse were analysed as a function of the three fixed effects: number of days after insemination, stimulus intensity, and treatment group. Two queens belonging to the seminal fluid treatment were excluded from the ocelli dataset because their measurements represented clear outliers due to small signal sizes and large technical noise. Factors or interaction-terms were added stepwise and $\chi^2$ significance values were obtained by comparing nested models (R function ANOVA). Only variables with $p<0.05$ were retained in the final model and all reported *P* values were tested against the final model. All models were also graphically checked for consistency with model assumptions of normality and homogeneity of variances.

## Natural mating flight behaviour of artificially inseminated queens

To corroborate our findings from our previous gene expression and electroretinogram experiments, we bred three rounds of 12 virgin sister queens and artificially inseminated them 8 days after hatching with either 6 µl of semen, Hayes saline or seminal fluid (four queens per treatment in each round, 12 queens per treatment, 36 queens in total). We sedated queens on ice, because previous studies showed that using $CO_2$ reduces the likelihood of queens embarking on mating flights (*Kocher et al., 2010*; *Niño et al., 2011*; *Niño et al., 2013a*). We fitted each queens with a RFID tag (mic3-TAG 64-bit RO), and re-introduced queens individually into queen-less nucleus hives. In a set-up that we previously used to monitor honeybee behaviour (*Dosselli et al., 2016*), we narrowed the hive entrances and forced individual queens to pass through a set of two RFID tag readers (iID MAJA module 4.1) when they were leaving or returning to their hives to participate in mating flights. This set-up allowed us to reliably monitor the flight behaviour of queens as queens leaving their hive would trigger the inner reader closer to the entrance before the outer reader closer to the exit, while returning queens would trigger the readers in the opposite order. Raw data recoded by the readers were collected in XML format on a SD memory card in the database box (iID HOST type MAJA 4.1) from where they were downloaded to a PC and assembled in a MySQL database. To identify potential mating flights we: (i) only evaluated complete sequences of reader recordings (in – out - out – in), (ii) only evaluated data recorded after 12:00 noon because honeybee queens only fly out in the afternoon to mate (*Winston, 1987*), and (iii) only retained data for flights of 406 s or longer, because the shortest average mating flight of a Hayes-inseminated queen that later laid eggs in our study was 406 s, and because similar thresholds have been applied in previous studies (*Heidinger et al., 2014*; *Dosselli et al., 2016*). We also recorded a queen's number of flights per day and her total flight duration. Two queens were discarded from subsequent analyses because the readers did not record any completed afternoon flight for them. This experiment was conducted at the University of Western Australia (31° 59' 5.143'' S, 115° 49' 15.553'' E) from the end of January to the end of March 2017. All hives were checked after the experiment for the presence of newly laid eggs and brood. To statistically compare effects in our field experiment we used IBM, SPSS version 23 for Mac.

## Acknowledgements

We thank Tiffane Bates for help with the breeding of bee material and for excellent general assistance in the apiary. This work was supported by the facilities of the Australian Research Council Centre of Excellence Program [CE140100008] and funded by the University of California Riverside, Australian Research Council grants [LP100100438, DP130100087, LP130100029] and ARC Future Fellowship [FT110100105] to BB, ARC Future Fellowship [FT110100528] to JMH, and an ERC Advanced Grant [323085] to JJB.

## Additional information

### Funding

| Funder | Grant reference number | Author |
|---|---|---|
| Australian Research Council | LP100100438 | Boris Baer |
| Australian Research Council | DP130100087 | Boris Baer |
| Australian Research Council | LP130100029 | Boris Baer |
| Australian Research Council | Future Fellowship FT110100105 | Boris Baer |
| University of California, Riverside | Faculty start up fund | Boris Baer |
| Australian Research Council | Future Fellowship FT110100528 | Jan M Hemmi |
| European Research Council | Advanced Grant 323085 | Jacobus J Boomsma |

The funders had no role in study design, data collection and interpretation, or the decision to submit the work for publication.

### Author contributions

Joanito Liberti, Conceptualization, Formal analysis, Investigation, Visualization, Writing—original draft, Project administration, Writing—review and editing; Julia Görner, Formal analysis, Investigation; Mat Welch, Conceptualization, Investigation, Writing—review and editing; Ryan Dosselli, Morten Schiøtt, Formal analysis, Writing—review and editing; Yuri Ogawa, Conceptualization, Formal analysis, Investigation, Writing—review and editing; Ian Castleden, Data curation, Formal analysis; Jan M Hemmi, Conceptualization, Formal analysis, Writing—review and editing; Barbara Baer-Imhoof, Formal analysis, Investigation, Writing—review and editing; Jacobus J Boomsma, Conceptualization, Supervision, Funding acquisition, Writing—original draft, Project administration, Writing—review and editing; Boris Baer, Conceptualization, Supervision, Funding acquisition, Investigation, Writing—original draft, Project administration, Writing—review and editing

### Author ORCIDs

Joanito Liberti (iD) https://orcid.org/0000-0002-4158-2591
Julia Görner (iD) https://orcid.org/0000-0003-4026-9197
Ryan Dosselli (iD) https://orcid.org/0000-0001-6524-5138
Morten Schiøtt (iD) https://orcid.org/0000-0002-4309-8090
Jan M Hemmi (iD) https://orcid.org/0000-0003-4629-9362
Jacobus J Boomsma (iD) https://orcid.org/0000-0002-3598-1609
Boris Baer (iD) https://orcid.org/0000-0002-1136-5967

### Decision letter and Author response

Decision letter https://doi.org/10.7554/eLife.45009.043
Author response https://doi.org/10.7554/eLife.45009.044

## Additional files

### Supplementary files

• Supplementary file 1. Summary of results from differential expression analyses performed with DESeq2 (*Love et al., 2014*). The total number of DEGs identified, the number of these DEGs that were up-regulated, and the number of DEGs that were down-regulated are reported in separate columns for each of the pair-wise comparisons between treatment groups in both RNA-sequencing experiments.
DOI: https://doi.org/10.7554/eLife.45009.023

• Supplementary file 2. Results of differential expression analyses performed with DESeq2 (*Love et al., 2014*). Differentially expressed genes at FDR < 0.1 are highlighted in yellow. Results from different pair-wise comparisons are reported in separate sheets.
DOI: https://doi.org/10.7554/eLife.45009.024

• Supplementary file 3. Summary of enriched Biological Process and Molecular Function GO terms at p<0.05 in all pair-wise comparisons between treatment groups in both RNA-seq experiments. Different pair-wise comparisons are reported in separate sheets.
DOI: https://doi.org/10.7554/eLife.45009.025

• Supplementary file 4. Summary of GAGE pathway analyses results mapping the DEG lists of each of our pair-wise comparisons to known metabolic and signaling KEGG pathways. Significant pathways (*q* values < 0.2) are presented in bold. Different pair-wise comparisons are reported in separate sheets.
DOI: https://doi.org/10.7554/eLife.45009.026

• Supplementary file 5. Annotation of the 37 genes showing consistent changes in expression between all pair-wise comparisons of semen and seminal fluid insemination treatments against controls.
DOI: https://doi.org/10.7554/eLife.45009.027

• Supplementary file 6. Results of analyses of overlap (Hypergeometric tests) in DEG lists between our study and *Manfredini et al. (2015)*. Overlap between each of our pair-wise comparison and the comparisons mated *versus* virgin and mated *versus* CO2-treated, and the overall overlap of all the DEGs identified in our study and in *Manfredini et al. (2015)* are reported in two separate tables. For each of our DEG lists, tables report the number of DEGs identified in our study, the number of shared DEGs with the specific comparisons by *Manfredini et al. (2015)*, and the representation factor and *P* value of the Hypergeometric test.
DOI: https://doi.org/10.7554/eLife.45009.028

• Supplementary file 7. Summary of enriched Biological Process and Molecular Function GO terms at p<0.05 for the DEGs shared with the *Manfredini et al. (2015)* study of naturally-inseminated honeybee queens.
DOI: https://doi.org/10.7554/eLife.45009.029

• Supplementary file 8. Results of a linear mixed effects model for flicker response amplitude of compound eyes, showing the significance of the fixed effects and their interactions. Significant effects (p<0.05) are reported in bold. df = degrees of freedom, $\chi^2$=chi squared statistic. The final model is shown below the table.
DOI: https://doi.org/10.7554/eLife.45009.030

• Supplementary file 9. Results of a linear mixed effects model for flicker response amplitude of compound eyes comparing the measurements taken on the second day post-inseminations for the nine queens that had exclusively been measured at day two with those of the 18 queens that were measured both on day 1 and 2. df = degrees of freedom, $\chi^2$=chi squared statistic. The final model is shown below the table.
DOI: https://doi.org/10.7554/eLife.45009.031

• Supplementary file 10. Results of a linear mixed effects model for flicker response amplitude of compound eyes after excluding all semen measurements, showing the significance of the fixed effects and their interactions. Significant effects (p<0.05) are reported in bold. df = degrees of freedom, $\chi^2$=chi squared statistic. The final model is shown below the table.
DOI: https://doi.org/10.7554/eLife.45009.032

• Supplementary file 11. Linear mixed effect model for flicker response amplitude of ocelli, showing significant factors and their interactions. Significant effects (p<0.05) are reported in bold. df = degrees of freedom, $\chi^2$=chi squared statistic. The final model is shown below the table.
DOI: https://doi.org/10.7554/eLife.45009.033

• Supplementary file 12. Linear mixed effect model for flicker response amplitude of ocelli after exclusion of semen measurements, showing significant factors and their interactions. Significant effects (p<0.05) are reported in bold. df = degrees of freedom, $\chi^2$=chi squared statistic. The final model is shown below the table.
DOI: https://doi.org/10.7554/eLife.45009.034

• Supplementary file 13. Results of a linear mixed effects model for contrast sensitivity of compound eyes, showing the significance of the fixed effects and their interactions. Significant effects (p<0.05) are reported in bold. df = degrees of freedom, $\chi^2$=chi squared statistic. The final model is shown below the table.
DOI: https://doi.org/10.7554/eLife.45009.035

• Supplementary file 14. Results of a linear mixed effects model for contrast sensitivity of compound eyes after exclusion of semen measurements, showing the significance of the fixed effects and their interactions. Significant effects (p<0.05) are reported in bold. df = degrees of freedom, $\chi^2$=chi squared statistic. The final model is shown below the table.
DOI: https://doi.org/10.7554/eLife.45009.036

• Supplementary file 15. RNA-sequencing statistics. For each sample the table shows the total number of reads obtained from the sequencer, the number of reads retained and discarded after the filtering steps (Trimmomatic and SortMeRNA), and the number or reads mapped and unmapped after reads were aligned to the honeybee genome using STAR.
DOI: https://doi.org/10.7554/eLife.45009.037

• Source code 1. Custom-made MATLAB software used to control ERG recordings.
DOI: https://doi.org/10.7554/eLife.45009.038

• Transparent reporting form
DOI: https://doi.org/10.7554/eLife.45009.039

## Data availability

RNA-sequencing data have been deposited in NCBI's Gene Expression Omnibus and are accessible through GEO Series accession number GSE127185 (https://www.ncbi.nlm.nih.gov/geo/query/acc.cgi?acc=GSE127185).

The following dataset was generated:

| Author(s) | Year | Dataset title | Dataset URL | Database and Identifier |
|---|---|---|---|---|
| Liberti J, Görner J, Welch M, Dosselli R, Schiøtt M, Ogawa Y, Castleden I, Hemmi JM, Baer-Imhoof B, Boomsma JJ, Baer B | 2019 | Seminal fluid compromises visual perception in honeybee queens reducing their survival during additional mating flights | https://www.ncbi.nlm.nih.gov/geo/query/acc.cgi?acc=GSE127185 | NCBI Gene Expression Omnibus, GSE127185 |

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
