## [Decision Letter]

Thank you for submitting your article "Seminal fluid compromises visual perception in honeybee queens reducing their survival during additional mating flights" for consideration by *eLife*. Your article has been reviewed by three peer reviewers, and the evaluation has been overseen by a Reviewing Editor and Ian Baldwin as the Senior Editor. The reviewers have opted to remain anonymous.

The reviewers have discussed the reviews with one another and the Reviewing Editor has drafted this decision to help you prepare a revised submission.

All three reviewers consider your study an interesting and novel contribution on the effects of seminal fluid on transcriptional responses in the brain of honeybees and the consequences for visual perception. Yet, all three reviewers are concerned about the assumption that what you found reflects sexual conflict. This conclusion needs to be toned down.

The reviewers have carefully read the manuscript and have identified a number of issues that need to be addressed to make the study suitable for publication in *eLife*. The main issues are summarized here.

The reviewers have concerns about the DEGs and their biological meaning and/or their relationship to what was seen in other studies. Please attend to this.

There are issues about the "getting lost" and whether/how this would necessarily only be vision-related. This will need your attention as well and could affect their overall conclusions.

The effects of carbon dioxide narcosis on bee behaviour should be included to place the results in context.

Finally, a number of instances have been identified where you did not fully include the context in the field – not citing references and work that is essential to include. Important earlier *Drosophila* work as well as honeybee work should be referenced in the manuscript to place the work in the relevant context.

Addressing these and the other comments by the reviewers will make the paper more accurate and thus more robust.

*Reviewer #1:*

This manuscript provides an interesting study of the consequences of insemination of honeybee queens on transcriptional responses in the brain including the phototransduction pathway and behavioural consequences. The text is well written and the experiments straightforward.

In the Results section the text refers to another study on brain transcriptional responses to insemination, albeit at 48 hours after insemination instead of the 24 hour timepoint used here. The authors compare the overlap in number of DEGs (12-15%) but do not tell whether the phototransduction pathways are altered in a way similar to what is recorded in the present study. That information is crucial for the present study. Please add this to the Results section or the Discussion section. This refers to the data in Figure 2 and Figure 3. Given that Manfredini et al., 2015 had already assessed the effects of natural insemination on brain transcriptomics, what was the reason for assessing the 24 hour time point in the present study and not the 48 hour time point?

The discussion on the sexual conflict is not clear enough to me. For example, in the Discussion section it is stated that the reduced visual capabilities are in the interest of the drones but given that queens carrying their sperm have a reduced probability of returning to the hive after a subsequent mating flight clearly indicates a considerable cost related to the effect of the seminal fluid. The Discussion and the Abstract suggest that the reduced visual capabilities are the target of the male's seminal fluid. However, given the tremendous cost (60% not returning, vs only 10% not returning in the control treatment – Figure 5). It is difficult to see this as part of a strategy of the males. Why could this not be the cost of using the seminal fluid to manipulate another, more rewarding, physiological process in the queen? This option has not been included in the discussion yet.

*Reviewer #2:*

Liberti et al. report findings consistent with the interesting hypothesis that molecules in the seminal fluid of honeybees impair the visual acuity of queens, hampering the latters' ability on subsequent mating flights and leading to lower survival. The results are considered and discussed from the perspective of sexual conflict, suggesting a strategy by which males permit some polyandry since they benefit from it, but not too much, since that would hamper their larger success.

The authors report three sets of findings. First, brain gene expression in honeybee females that were inseminated with semen or seminal fluid showed differences to that of females inseminated with control (saline or no fluid). Among DEGs were some in genes associated with vision in honeybees or *Drosophila*. Second, ERG measurements of visual acuity were compared between females inseminated with semen or seminal fluid or controls. The data suggested that seminal fluid in particular impaired visual acuity, especially in eyes; less in ocelli. The timing of the effect paralleled when honeybee females take their subsequent mating flights. Third, the likelihood of female honeybees failing to return to hives following those flights was lower following semen or seminal fluid insemination, and such inseminated females also appeared to spend more time around the hive opening suggesting that they were less likely to fly off.

The experiments seem designed well, and conducted with care and with proper controls, sample sizes, replicates. The data were analyzed by appropriate methods. The writing is clear, though a bit repetitive in the Discussion section. The novel hypothesis that is proposed will interest a broad range of biologists, once it can be made less speculative. Attention to comments 1-3, in particular, will help that.

1) The data that shows gene expression differences in brains of queens inseminated with various fluids seems robust in a technical sense, but:a) The authors mention that several studies (Kocher et al., 2008; Kocher, Tarpy and Grozinger, 2010; Manfredini et al., 2015) examined brain gene expression in naturally-inseminated queens, but did not give detailed comparisons of their findings with those of the previous studies. Were the 12-14% of shared DEGs between their study and Manfredini et al., 2015 inclusive of the vision-related ones? Enriched for those? Depleted for them?

b) Some genes shown in Figure 2 are expressed in eyes, but Liberti et al. examined only brains. So, the relevance is unclear for including, and noting changes in, eye genes.

2) Many genes detected, including those in Figure 2, are needed for other behaviors and some such as Actin and Cam, for overall health and viability. Thus, effects of seminal fluid on female honeybees may be more general, with the diminished vision shown by the ERGs simply being one effect of decreased vitality. From this perspective, it was surprising that Liberti et al. did not consider work in *Drosophila* that showed that seminal fluid contains molecules that decrease female survival (Chapman, et al., 1995) and that the sex peptide, mentioned by Liberti et al. for other contexts, contributes to this (Wigby and Chapman, 2005), though is likely not its sole basis. Please add this dimension, and citations, to the paper.

3) Related to comment 2, the authors interpret the lack of return of inseminated females after the later mating flight to their being "more likely to get lost" because of diminished vision. That is certainly a possibility but it seems equally likely that the females' health or survival were negatively impacted by seminal fluid, making them less possessed of the energy to return, even without being "lost". Relatedly, the authors interpret the inseminated females' tendency to spend more time at the hive entrance as due to their being "disoriented or distressed by sunlight". Again, there could instead be an energy- or health-based explanation. Unless the authors can document that the females were getting lost, or were disoriented, the text needs to be revised to consider this possibility and to not interpret the flight experiments solely in terms of a vision-related cause.

4) In some places in the paper, the authors compare the effects of whole semen to that of seminal fluid, but in others, such as subsection “Analyses of gene expression in queen brains”, they don't. Please discuss whether there are differences in the transcriptome changes that can be attributed to sperm or seminal fluid alone, as you did for vision.

5) Can the authors provide any transcriptome-level basis for the findings presented in subsection “Visual perception of queens after experimental exposure to seminal fluid”? Including, perhaps, qPCR of some of the critical genes at several time points, to confirm the temporal changes that they discuss?

6) Some of the arguments made by the authors about the basis and effects of sexual conflicts have previously been made about the rapid DNA-sequence evolution of seminal protein genes in *Drosophila*, and other organisms. It would broaden the appeal of this paper to mention those studies as part of the context in which you consider and interpret your results.

*Reviewer #3:*

This paper suggests that male honey bees manipulate queen mating behaviour via their seminal fluid. The authors hypothesise that this is a manifestation of inter-sexual conflict, in which males are selected to reduce polyandry and queens to increase it. It is an exciting, original contribution. Because of its novelty, it will need a great deal of corroboration, but I support publication as a magnificent first step in what will likely be a long journey.

I do think that the authors should be more measured in their claim that changes in gene expression are a manifestation of sexual conflict. There may not be a reduction in visual acuity but only a reduction in phototaxis. Agreed that the failure of queens that had been inseminated with semen/seminal fluid to return from mating flights is suggestive of sexual conflict.

The insemination procedure used here (in the main experiment) used narcosis with carbon dioxide to immobilize the queen. Carbon dioxide narcosis alone, without insemination, semen, or a mating flight, initiates oviposition in queens, is accompanied by all the behavioural and physiological changes that follow natural mating, including cessation of mating flights. This experiment held CO_2_ narcosis constant and compared the brain transcriptomes of queens that received seminal fluid/whole semen and Haynes solution. But any effects of treatment are secondary to the CO_2_ narcosis. It is remarkable that this approach found any affects at all. Nonetheless, the results seem convincing. Genes related to vision are differentially expressed between queens exposed to seminal fluid and those that were not, and responses to light were also affected.

Although several studies have reported post-insemination changes in expression of genes related to vision and light sensitivity, I find this a surprising mechanism to control the urge to mate. Of the 12 or so Apis species, about half nest in the open. Queens of the open-nesting species are exposed to near-ambient light throughout life. So, although reduced attraction to light might be a plausible mechanism to keep the queens of cavity-nesting species at home, it can't be the ancestral mechanism. This matter should be discussed in the fifth paragraph of the Discussion section.

A surprising finding is that queens exposed to semen were more likely to get lost than queens that were sham inseminated (subsection “Mating flight behaviour after experimental seminal fluid exposure”, Discussion section). I would have thought that if a queen was so adversely affected by semen that her vision was impaired, she would stay in her hive and lay eggs. 65% of 11 semen-inseminated queens were lost in this experiment (Figure 5). My experience is that only 5% of inseminated queens do not survive to lay eggs. The difference is that standard procedure for instrumentally inseminated queens is to confine them to their hives until they lay. This point should be discussed.

The following two papers, which came to diametrically opposed conclusions) are relevant to this manuscript and should be discussed: Schlüns et al., (2005); Tarpy and Page, (2000).

---

## [Author Response]

All three reviewers consider your study an interesting and novel contribution on the effects of seminal fluid on transcriptional responses in the brain of honeybees and the consequences for visual perception. Yet, all three reviewers are concerned about the assumption that what you found reflects sexual conflict. This conclusion needs to be toned down.The reviewers have carefully read the manuscript and have identified a number of issues that need to be addressed to make the study suitable for publication in eLife. The main issues are summarized here.The reviewers have concerns about the DEGs and their biological meaning and/or their relationship to what was seen in other studies. Please attend to this.

We have now performed additional bioinformatics comparisons using the data from the Manfredini et al., 2015 study to quantify the degree of overlap in vision-associated genes. We confirm the overlap in several vision-associated genes, including several key genes of the phototransduction pathway. We can therefore conclude that our experimental set up using artificial insemination generated similar responses in queen vision as those induced by natural mating. To illustrate this, of the eight vision-associated genes listed in the Additional File 6 of Manfredini et al., 2015, four (*ninaC, ninaA, norpA, chp*) were DEGs in our study, which represents a statistically significant overlap (Hypergeometric test, representation factor = 5.57, *P =* 0.0002). A fifth gene (*Arr2*) in the Manfredini study was only found in the seminal fluid vs. semen comparison of our second RNA-seq experiment and thus represents a partial replication. We furthermore analysed the DEG datasets of Manfredini et al., 2015 to more comprehensively assess overlap in additional genes implicated in vision that may not have been reported in their Additional File 6. This showed that another six vision-associated DEGs were shared between both studies (*Actin (a), Actin (b), GNB1, TRP, myosin-IIIb, Hemicentin-1-like*), the first five of which are important components of the phototransduction pathway and the latter one was the gene which consistently had the highest fold-change across our pair-wise comparisons between semen or seminal fluid insemination treatments and controls. This gene has been implicated in age-related macular degeneration in humans (a progressive degeneration of photoreceptors), although it is unclear whether it plays a role in similar degeneration of the retina in insects. We have now added these additional analyses to the Results section.

We have also added text to the Discussion section presenting further evidence of previous studies that found an effect of insemination on the phototransduction pathway in the bumblebee *Bombus terrestris* and the egg parasitoid wasp *Anastatus disparis* (Manfredini et al., 2017; Liu and Hao 2019). In both species, males enforce female monogamy through components of their ejaculates, but multiple mating can sometimes occur. More specifically, the Manfredini et al., 2017 study found remarkably similar effects of insemination on pathways in bumblebee queens as the ones we now identify in the honeybee, including phototransduction, neuroactive ligand-receptor interactions, the Hippo signaling pathway and the phagosome pathway, the latter two of which we found to be significantly affected in comparisons between seminal fluid and Hayes saline but which we did not properly discuss in the previous version of our manuscript. It thus appears that homologous pathways are involved and that they are affected in similar ways by insemination across social and non-social Hymenoptera.

The reviewers also raised questions about our interpretation of what these effects imply in terms of sexual conflict. We believe that this skepticism is related to not fully appreciating that honeybees and other advanced social Hymenoptera have mating systems that are fundamentally different from those present in flies such as *Drosophila* or other non-social experimental model systems. This is because multiple insemination of queens does not imply promiscuity in the usual sense of re-mating later in life. This is a fundamental difference that simplifies the interaction dynamics between the sexes to such degree that precise predictions about selection regimes are possible. Our sexual conflict interpretation therefore has a firm foundation in natural history knowledge of social insects and is consistent with conceptual expectations that have been presented in review papers for more than a decade (Boomsma, Baer and Heinze, 2005; Heinze and Schrempf 2008; Boomsma, 2009; Boomsma, 2013. To address the reviewers’ concerns and make our rationale more transparent we now:

1) Spell out in more detail why the lack of re-mating later in life and the need to produce many sterile workers before a colony can produce dispersing virgin queens imply that negative effects of insemination on general female health will not be selected for. This prediction is consistent with all empirical evidence in social insects so far and implies that the expression of sexual conflict mediated by reproductive fluids between honeybees and other polyandrous social insects and fruit files and other non-social insects are fundamentally different.

2) Spell out in more detail how different the honeybee mating system is from the mating systems of other polyandrous social insects, because honeybee queens engage in subsequent mating flights over several days, while all other social insect queens obtain their life-time complement of sperm on a single day. We can therefore predict that ejaculates of honeybee drones have been under selection to risk the lives of queens during the mating period of up to 4 days (Winston, 1987). However, these same seminal-fluid-induced manipulations should not cause lasting health effects in surviving queens, as documented in promiscuous fruit flies. This hypothesis drove our study and all our results are consistent with this logic.

In sum, we now explain much better that detailed knowledge of the natural history of polyandrous social insects implies that the logic of our predicted sexual conflict scenario is simple and parsimonious, and we show that our study generated consistent qualitative support for the predictions that we made. At the same time, we now are much more explicit about the quantitative details that remained beyond the scope of our present study and how many of these offer exciting opportunities for follow-up research to unravel proximate causations.

There are issues about the "getting lost" and whether/how this would necessarily only be vision-related. This will need your attention as well and could affect their overall conclusions.

To address this point, we substantially restructured the Discussion section and clarified our rationale as already lined out in our responses to the previous point. We made explicit that further research will be required to assess whether a causal link exists between our genetic, physiological and behavioural experiments. We also added text to explain why the mating systems of social Hymenoptera would never produce selection for male fitness-enhancing traits that negatively affect queen physiological performance in the longer run as has been observed in other insects where females continue to re-mate later in life (e.g. *Drosophila*; Chapman et al., 1995; Wigby and Chapman, 2005). The key issues are that:

1) In all social insects other than honeybees, where a complete filling of spermathecae is achieved during a single day, additional sexual activity is pointless. This is different in honeybees, where the queen spermatheca is filled with semen over several days and often via inseminations obtained during subsequent mating flights.

2) All queens of advanced social insect species need to produce multiple cohorts of sterile workers before their colony is large enough to produce sexuals that pass on genes to future generations. This peculiar set of life-history traits, unique to social insects, makes it highly unlikely that any trait that reduces queen survival or female lifetime fecundity can evolve (Boomsma, 2013; Heinze and Schrempf, 2008). These theoretical predictions are supported by experimental work on ants and bees (e.g. Schrempf et al., 2005; Tsuji et al., 1996; Lopez-Vaamonde et al., 2009), showing that insemination typically increases queen lifespan rather than decreasing it (Tsuji et al., 1996; Schrempf et al., 2005; Lopez-Vaamonde, 2009; Rueppell et al., 2015), likely via beneficial effects of cooperative seminal fluid proteins (Fuessl et al., 2014; Fuessl et al., 2018).

The “getting lost” effect makes sense when one realizes how high the reproductive fitness stakes are for drones already present as ejaculates within a queen. They are all ‘in the same boat’ towards future failure or joint reproductive success. In such a situation, a collective death risk of manipulation would be maintained by selection provided that on average the number of queens that fail to locate a drone congregation and successfully mate would be higher than the number of queens who get lost and die, as this would result in a fitness increase for first inseminating males compared to a scenario without any sexual conflict. We are fully aware that our apiary experiment only offers a qualitative test of this logic because we only measured the potential cost of this manipulation. We remain convinced that these more detailed explanations are sound and hope they now clarify why the data that we obtained in our apiary experiment make perfect sense because they match our predictions even though quantitatively our experimental treatments may have imposed unnaturally high levels of male manipulation and thus have induced more queen mortality that would likely happen in natural mating flights.

The effects of carbon dioxide narcosis on bee behaviour should be included to place the results in context.

We appreciate this comment raised by reviewer 3 and we have now added information on the physiological effects induced by CO_2_ narcosis both in the Introduction and in the Discussion section. We now explain how CO_2_ can produce some of the mating-induced changes in physiology, but that nevertheless by keeping CO_2_ narcosis constant across both our RNA-seq and ERG experiments (and matching controls) we were still able to recover the specific effects of seminal fluid on the brain transcriptome of queens.

Finally, a number of instances have been identified where you did not fully include the context in the field – not citing references and work that is essential to include. Important earlier *Drosophila* work as well as honeybee work should be referenced in the manuscript to place the work in the relevant context.

This point is related to issues raised above. *Drosophila* experiments are indeed highly relevant although – as we now explain more clearly – not fully comparable for every aspect of our study. We have now added additional citations to relevant earlier publications that investigated the *Drosophila* sex peptide and the rapid evolution of seminal fluid proteins (e.g. Swanson and Vacquier, 2002; Haerty et al., 2007). We also substantially expanded our reference list to justify our additions to the Discussion section in response to the issues raised above. These include the functional implications of the genes and pathways we uncovered in queen brains (e.g. Halder and Johnson, 2011; Stuart and Ezekowitz, 2005), the effects of CO_2_ exposure of honeybee queens (e.g. Vergoz et al., 2012), the post-mating effects on phototransduction in other insects (e.g. Manfredini et al., 2017; Liu and Hao, 2019), the documented increases of queen longevity in social insects rather than a decrease as is usually observed in promiscuous mating systems where male seminal fluid can harm female condition (e.g. Schrempf et al., 2005; Lopez-Vaamonde, 2009), and the known natural history and proximate/ultimate causation of repeated mating flights by honeybee queens (e.g. Tarpy and Page, 2000; Schlüns et al., 2005).

Reviewer #1:This manuscript provides an interesting study of the consequences of insemination of honeybee queens on transcriptional responses in the brain including the phototransduction pathway and behavioural consequences. The text is well written and the experiments straightforward.In the Results section the text refers to another study on brain transcriptional responses to insemination, albeit at 48 hours after insemination instead of the 24 hour timepoint used here. The authors compare the overlap in number of DEGs (12-15%) but do not tell whether the phototransduction pathways are altered in a way similar to what is recorded in the present study. That information is crucial for the present study. Please add this to the Results section or the Discussion section. This refers to the data in Figure 2 and Figure 3. Given that Manfredini et al., 2015 had already assessed the effects of natural insemination on brain transcriptomics, what was the reason for assessing the 24hour time point in the present study and not the 48 hour time point?

As already pointed out in our responses to the Editor’s comments, we now provide more detailed comparisons to assess the degree of overlap in vision-related genes with Manfredini et al., 2015.

We also added evidence from recent studies showing that other Hymenoptera (bumblebees and an egg parasitoid wasp) in which males enforce female monogamy documented similar effects on phototransduction pathway in the brain following mating.

In the previous and current version of our manuscript, we compared our study with Manfredini et al., 2015 only to verify a posteriori that the effects we observed matched those found after natural insemination. As we point out, this was not the main aim of our study because we were primarily interested to link our previous research on seminal fluid components to physiological and phenotypic changes in queens after they were exposed to male reproductive fluids. For the initial gene expression experiments, we prioritized sampling queens 24 hours after they had been artificially inseminated rather than 48 hours, the time-point following natural insemination in Manfredini et al., 2015, because 24 hours represents the critical first opportunity for queens to naturally fly out again to accumulate additional inseminations. At this point in time it is most obvious that pre-storage sperm competition is still operating because life-time sperm storage in the honeybee can take more than 40 hours (Woyke, 1983). Our experimental setup thus reflected our estimate of when the potential for sexual antagonism between competing ejaculates and the inseminated queen would likely be maximally expressed.

The discussion on the sexual conflict is not clear enough to me. For example, in the Discussion section it is stated that the reduced visual capabilities are in the interest of the drones but given that queens carrying their sperm have a reduced probability of returning to the hive after a subsequent mating flight clearly indicates a considerable cost related to the effect of the seminal fluid. The Discussion and the Abstract suggest that the reduced visual capabilities are the target of the male's seminal fluid. However, given the tremendous cost (60% not returning, vs only 10% not returning in the control treatment – Figure 5). It is difficult to see this as part of a strategy of the males. Why could this not be the cost of using the seminal fluid to manipulate another, more rewarding, physiological process in the queen? This option has not been included in the discussion yet.

As already explained in our responses to the Editor, we have now modified and expanded our text in the Discussion section to better explain the rationale of our sexual conflict reasoning:

i) The balance between male manipulation and female counter-measures is likely to be more even under natural circumstances than the ones we imposed in our experiment, because genetic variation in semen and seminal fluid used in the experimental treatments of our study (pools of many drones, hundreds in the case of the pure seminal fluid treatment) was likely higher than what average naturally inseminated queens experience during their first mating flight. The treatments and controls of our experiment were chosen such as to generate maximal differences, not to necessarily emulate the natural conditions. We have now modified the relevant sections in the manuscript to improve the clarity of our rationale in subsection “Reductions of female visual perception after mating” and subsection “Sexual conflict over the number of mating flights”. This includes an evaluation of the cost of this conflict for manipulating males, for whom some modest risk of mortality is expected to be worth taking if the benefit of higher paternity is large enough. Further empirical work is needed to test whether already inseminated queens indeed have reduced capability of locating drone congregation areas. We now make several of these issues clearer in subsection “Further considerations, caveats and suggestions for future research”.

ii) As already explained above, it is important to appreciate that social insects provide unique research opportunities because any negative effects on the physiological performance of an inseminated queen induced by insemination will not be favored by natural selection. A male effect like this would only be conceivable if the expression of such negative effects on the queen physiology occur very late in the life of a queen, i.e. when it would be comparable with senescence rather than immediate sexual conflict. This contrasts with other animals with continuous re-mating opportunities, such as fruit flies, where insemination has indeed been shown to generally reduce female fitness later in life. That is why we predicted that any manipulative effect should be restricted to the timeframe of the actual mating period (a few days) and should only target the performance of queens during the mating flights. The important result of our study is that we find that the negative effect of seminal fluid on queen eyesight does indeed appear to handicap queens in returning safely to the hive after an additional flight, which may also imply reduced abilities in finding a drone aggregation. We now explain in more detail what factors may have made our apiary experiment quantitatively inaccurate, but without compromising its general validity in subsection “Further considerations, caveats and suggestions for future research”.

iii) In an evolutionary perspective, average tendencies of queens to embark on multiple flights result from a balanced compromise between benefits (increased genetic diversity for their colonies) and costs (predation risk and additional survival costs imposed by male manipulation) of additional flights. Hence, these survival costs likely reduce queen re-mating tendencies. Furthermore, the visual handicap resulting in some queens losing their way back to the hive may also reflect the way queens navigate to locate the congregation areas in which drones wait for the arrival of queens to mate with. The manipulation of visual perception would therefore result in a substantial reduction of mating success in queen flights following the first flight. We could not directly measure whether such direct fitness benefits for males exist as this is technically very challenging, but we believe that, contrary to what the reviewer stated here, the survival cost for queens can easily be interpreted as part of a male strategy to reduce queen promiscuity. Note also that we did not find any effect on gene expression that would underlie other physiological queen traits that males could manipulate (and do manipulate in *Drosophila*) than the highly consistent effects on the phototransduction pathway that we identified.

Reviewer #2:[…] The experiments seem designed well, and conducted with care and with proper controls, sample sizes, replicates. The data were analyzed by appropriate methods. The writing is clear, though a bit repetitive in the Discussion section. The novel hypothesis that is proposed will interest a broad range of biologists, once it can be made less speculative. Attention to comments 1-3, in particular, will help that.1) The data that show gene expression differences in brains of queens inseminated with various fluids seem robust in a technical sense, but:a) The authors mention that several studies (Kocher et al., 2008; Kocher, Tarpy and Grozinger, 2010; Manfredini et al., 2015) examined brain gene expression in naturally-inseminated queens, but did not give detailed comparisons of their findings with those of the previous studies. Were the 12-14% of shared DEGs between their study and ref. 30 inclusive of the vision-related ones? Enriched for those? Depleted for them?b) Some genes shown in figure 2 are expressed in eyes, but Liberti et al. examined only brains. So, the relevance is unclear for including, and noting changes in, eye genes.

Given the similar comments by the other reviewers, we decided to address these points in our general response to the Editor above, which led to substantial modifications and additions of our text now that we compare our findings with previous work in more detail. Concerning the point raised here, we now report a remarkable overlap of neurogenomic signatures associated with post-mating changes in the honeybee and bumblebee (as reported by Manfredini et al., 2017), which also includes the phototransduction pathway. It is important to point out that a gene-level comparison between the DEGs in our study and those of the Manfredini bumblebee study is technically impossible because Manfredini et al., used Contig ids to describe their DEGs that cannot be readily converted into other gene ids. These contigs included several genes each and the authors did not provide the genome coordinates of the specific genes or a fasta file of the gene sequences. Without a link between annotations in their DEG tables, gene ids, and corresponding sequences it is impossible to establish which are the orthologous genes shared with the honeybee and thus to assess the overlap in specific DEGs between the studies.

The phototransduction genes that encode for the proteins shown in Figure 2 are expressed in photoreceptor cells in the eyes whose visual fibers extend in the optic lobe of the honeybee brain (Wernet et al., 2015; Ehmer and Gronenberg, 2002), where they carry the electrical signal generated from the light impulses detected by the retina. The KEGG model that we used to map the fold-changes in gene expression in our pair-wise comparisons represents a *Drosophila* rhabdomeric photoreceptor cell and also indicates where the cell body lies (see Figure 2). We therefore added “Light captured by eye” and “Transfer of signal to brain” to the left and right of each figure panel, respectively. We also added a sentence to the figure caption to explain that the visual fibers extend into the optic lobes of the brain.

2) Many genes detected, including those in figure 2, are needed for other behaviors and some such as Actin and Cam, for overall health and viability. Thus, effects of seminal fluid on female honeybees may be more general, with the diminished vision shown by the ERGs simply being one effect of decreased vitality. From this perspective, it was surprising that Liberti et al. did not consider work in *Drosophila* that showed that seminal fluid contains molecules that decrease female survival (Chapman, et al., 1995) and that the sex peptide, mentioned by Liberti et al. for other contexts, contributes to this (Wigby and Chapman, 2005), though is likely not its sole basis. Please add this dimension, and citations, to the paper.3) Related to comment 2, the authors interpret the lack of return of inseminated females after the later mating flight to their being "more likely to get lost" because of diminished vision. That is certainly a possibility, but it seems equally likely that the females' health or survival were negatively impacted by seminal fluid, making them less possessed of the energy to return, even without being "lost". Relatedly, the authors interpret the inseminated females' tendency to spend more time at the hive entrance as due to their being "disoriented or distressed by sunlight". Again, there could instead be an energy- or health-based explanation. Unless the authors can document that the females were getting lost, or were disoriented, the text needs to be revised to consider this possibility and to not interpret the flight experiments solely in terms of a vision-related cause.

It is important to remember that the genetic and phenotypic effects that we quantified were conducted in a comparable setup because we had appropriate control groups of queens. As the reviewer acknowledges, any of the phenotypic changes therefore had to be triggered by seminal fluid. The remaining question raised here is then whether the links between genomic changes and behavioural responses are causal or correlative. While several of the genes that we identified as being differentially expressed in queens exposed to seminal fluid also play roles in other pathways, the statistical methods that we employed specifically tested whether given pathways in a DEG dataset were enriched while simultaneously taking into account all the pathways that the focal genes are known to be involved in as well. We therefore believe that this analysis gave us a clear signal corrected for possible confounders. The pathways that stood out as having the most consistent perturbations were the phototransduction and the neuroactive ligand-receptor interaction pathways, but we also identified somewhat less consistent changes in the phagosome pathway, the Hippo signaling pathway and the tyrosine metabolism and ribosome pathways, which we now discuss in more detail. Apart from these statistical tests, the confidence that our DEG datasets suggest specific, and not by-product, effects of seminal fluid on the phototransduction pathway comes from multiple other studies that have shown vision-associated genes to be altered by natural insemination in the honeybee, in the bumblebee and in other insect species, as we have now outlined in detail in our general response to the Editor’s summary of the reviews. We believe that the cumulative evidence is consistent with insemination specifically affecting the visual transduction process and not queen health or long-term physiological function in a more general sense.

The second compelling reason for dismissing the possibility of generally negative health effects of seminal fluid in the honeybee or any other advanced social insect is that it is impossible to conceive of a selection regime that would maintain such harmful male effects. We already mentioned this in our responses above but detail the logic here as we have done in the revised manuscript. These negative health effects make sense in *Drosophila* where the sex peptide functions are well documented, because females mate promiscuously throughout their adult lives. Each male is therefore selected to maximise his own paternity and because there is no parental care all that counts is fertilised eggs. Later health costs for females and a lower likelihood of their future reproduction are therefore unimportant for every focal male’s fitness – they are just collateral damage from his perspective. This is fundamentally different across the social Hymenoptera because all ejaculates that a queen will store enter storage early in a queen’s life and before she starts to lay eggs. While this is expected to result in sperm competition for a very brief period prior to sperm storage, queens are expected to terminate sperm competition quickly, a phenomenon that we recently documented at the proteomic level in *Atta* leaf-cutting ants (Dosselli et al., 2018). Sperm competition also occurs in the honeybee but it is absent in singly inseminated bumblebees (den Boer at al., 2010), and must be terminated in a similar way because queens never re-mate later in life and would quickly lose their capacity to fertilise eggs if ejaculates would continue to compete. This is particularly so because queens always need to produce many fertilised eggs that will produce sterile workers first, before producing new queens when a colony has grown big enough (in honeybees this implies swarming). Any even remotely negative effect of seminal fluid on general queen health will therefore reduce male fitness to the same extent as queen fitness so it is impossible to imagine natural selection will make any such mutants increase in frequency. The details of these special mating system conditions have been outlined in a series of review papers (Boomsma et al., 2005; Boomsma, 2009, 2013), so we had merely summarized them in the present paper. We have now gone through that text again and added further detail to make these principles clearer – along the lines of what we specified in our general response to the Editor’s summary.

We agree, that future work should unravel the exact causal molecular links between the results that we obtained with RNA-sequencing, electroretinography and our behavioural trials (Discussion section), but such work – beyond the scope of our present study – will clarify the proximate mechanisms involved. It will not change the ultimate evolutionary logic predicting that generally negative effects on queen health should not have evolved – and such effects have indeed never been documented in social insects. To clarify this point we added references in the Discussion section that provide evidence that continuing female re-mating is expected to select for seminal fluid proteins that compromise female health and decrease later female survival (citing the literature the reviewer suggested), while at the same time contrasting this expectation with the opposite logic for social insects as summarized above (Discussion section).

4. In some places in the paper, the authors compare the effects of whole semen to that of seminal fluid, but in others, such as the first section of results, they don't. Please discuss whether there are differences in the transcriptome changes that can be attributed to sperm or seminal fluid alone, as you did for vision.

We agree that such pairwise comparisons are interesting in the context of this manuscript, The comparison between semen and seminal fluid only yielded a single DEG in the first RNA-seq experiment and consequently no GO term or pathway was significantly enriched. This was very different in the second RNA-seq experiment with a substantially higher detection power resulting in 802 DEGs, albeit with only subtle fold-changes (-1< log2 (fold change) < 1). We have now added text in lines 164-174 about the GO terms that were significantly enriched in the DEG list of the seminal fluid vs. semen comparison.

5) Can the authors provide any transcriptome-level basis for the findings presented in subsection “Visual perception of queens after experimental exposure to seminal fluid”? Including, perhaps, qPCR of some of the critical genes at several time points, to confirm the temporal changes that they discuss?

We agree that it would be nice to perform a longitudinal follow up study, with more time-points, also extending beyond day 2, to assess whether these neurogenomic effects are temporary and specific to the mating flight period or permanent or would persist longer. However, we hope the reviewer understands this would require a substantial amount of additional investigation beyond the scope of the present paper. Testing gene expression changes on the same set of queens that we employed in the ERG experiment was not possible for technical reasons.

6) Some of the arguments made by the authors about the basis and effects of sexual conflicts have previously been made about the rapid DNA-sequence evolution of seminal protein genes in *Drosophila*, and other organisms. It would broaden the appeal of this paper to mention those studies as part of the context in which you consider and interpret your results.

We agree and added this information and references to the Discussion section where we discuss our findings in relation to a putative arms-race dynamics between males targeting queen vision for impairment by seminal fluid and queens evolving counter adaptations to resist these male manipulations. However, such arms race dynamics in *Drosophila* would be fundamentally different than those of the honeybee. In fruit flies, the sexual antagonism is between how much exploitative damage males (seminal fluid) can get away with and how efficient female defences are before damage starts backfiring on a male’s own fitness. In the honeybee (and other social Hymenoptera with multiply inseminated queens) essentially all sexual conflict is terminated once queens have permanently stored ejaculates. In the honeybee, the putative arms race is about the rate of visual perception loss per unit of seminal fluid that drones can impose and that queens might be able to at least partly neutralize during the very few days in which the number of inseminations remains undecided. Both arms races are about sexual conflicts, but the arenas in which they play out are different. The general explanations that we have now added of the fundamental differences between social and non-social mating systems, and between honeybees and other polyandrous social insects (see our general response to the Editor’s summary above) should now have clarified our rationale.

Reviewer #3:This paper suggests that male honey bees manipulate queen mating behaviour via their seminal fluid. The authors hypothesise that this is a manifestation of inter-sexual conflict, in which males are selected to reduce polyandry and queens to increase it. It is an exciting, original contribution. Because of its novelty, it will need a great deal of corroboration, but I support publication as a magnificent first step in what will likely be a long journey.I do think that the authors should be more measured in their claim that changes in gene expression are a manifestation of sexual conflict. There may not be a reduction in visual acuity but only a reduction in phototaxis. Agreed that the failure of queens that had been inseminated with semen/seminal fluid to return from mating flights is suggestive of sexual conflict.

See also our responses above to the Editor and the other reviewers concerning this point. We already discussed the possibility that the effects we demonstrated should have downstream phototactic influences on queen behaviour a few days later when they start egg-laying and become confined within a dark hive environment. We have now extended this section to better clarify why a purely adaptive change in queen phototaxis makes little sense in light of our findings (Discussion section). We also acknowledge that more research is required to establish the causal link between the gene expression, physiological and behavioural changes we identified (Discussion section). We appreciate the idea that changes in phototaxis may represent an adaptive transition common to social Hymenoptera whose queens will remain confined in a dark environment to lay eggs, a transition that males appear to manipulate to their own benefit as a form of sensory exploitation. However, we also note that changes in phototaxis do not necessarily imply that photoreceptors are altered – although we can see that the opposite causation can work.

Nevertheless, following this and previous reviewer comments, we have reworded our Discussion section statements on the expression of sexual conflict and arms race dynamics so we separate more clearly between ultimate evolutionary logic leading to specific adaptations that would make evolutionary sense, and proximate causation on which our study is merely a first pioneering attempt.

The insemination procedure used here (in the main experiment) used narcosis with carbon dioxide to immobilize the queen. Carbon dioxide narcosis alone, without insemination, semen, or a mating flight, initiates oviposition in queens, is accompanied by all the behavioural and physiological changes that follow natural mating, including cessation of mating flights. This experiment held CO_2_ narcosis constant and compared the brain transcriptomes of queens that received seminal fluid/whole semen and Haynes solution. But any effects of treatment are secondary to the CO_2_ narcosis. It is remarkable that this approach found any affects at all. Nonetheless, the results seem convincing. Genes related to vision are differentially expressed between queens exposed to seminal fluid and those that were not, and responses to light were also affected.

We now report what is known about the effects of CO_2_ exposure on honeybee queens in the Introduction. All of the queens we used for our experiments were exposed to minimal amounts of CO_2_, i.e. durations of exposure were substantially shorter than those normally used during standard artificial insemination work of honeybee queens, where they receive a first narcosis lasting 3 to 10 minutes (as suggested by the reviewer in the minor comments below) and then a second exposure during AI on the following day. We also discuss in subsection “Further considerations, caveats and suggestions for future research” that the narcosis may have masked other changes induced by seminal fluid that could also be artificially induced by the CO_2_ narcosis. As the reviewer acknowledges, we nevertheless recovered relevant gene expression changes in phototransduction and other biological processes that are induced by seminal fluid.

Although several studies have reported post-insemination changes in expression of genes related to vision and light sensitivity, I find this a surprising mechanism to control the urge to mate. Of the 12 or so Apis species, about half nest in the open. Queens of the open-nesting species are exposed to near-ambient light throughout life. So, although reduced attraction to light might be a plausible mechanism to keep the queens of cavity-nesting species at home, it can't be the ancestral mechanism. This matter should be discussed in the fifth paragraph of the Discussion section.

What the reviewer points out here is that seminal fluid does only indirectly affect a queen’s decision to fly from the hive, but that the primary function of eyesight is to locate a drone aggregation and find the way back to the colony. The need to effectively locate a drone aggregation jointly applies to all honeybee species no matter whether they nest in the open or in cavities. However, in species nesting in the open, queens will normally not be at the light exposed surface because they are generally negatively phototactic unless they want to fly out to obtain inseminations or swarm. We agree with the reviewer that more studies will be needed to unravel proximate causation of the patterns that we uncovered. For example, semen- and seminal-fluid-inseminated queens seemed to spend more time at hive entrances than saline-inseminated queens, but they did not end up embarking on a lower number of flights than saline-inseminated queens, suggesting that instinctive drives build up over time and reluctance can be overcome. It would be interesting to know the causation pathways of this phenomenon, whether the strength of inhibition correlates with the amount of seminal fluid received, and if this strength varies across the different honeybee species depending on the kind of ‘equilibrium’ at which drone-queen arms races settle, something that could be clarified in future studies.

A well resolved phylogeny of the genus *Apis* would allow insight in whether cavity nesting or open nesting was ancestral and consequently offer an explicit framework for comparative studies. We note, however, that effects on phototransduction are not exclusive for the genus *Apis*, as they have been reported in the bumblebee (*Bombus terrestris*), in the egg parasitoid wasp *Anastatus disparis* and in *Drosophila* (Gioti et al., 2012; Manfredini et al., 2017; Liu and Hao 2019), a series of insect species in which males manipulate females to reduce their promiscuity rates (albeit in different time windows) across a variety of environmental light conditions in which these insects lay eggs. It thus appears that eyesight manipulations by males may be ancestral and that the common ancestor of the bumblebees and the honeybees likely had this inhibition even though almost all bumblebees appear to practice single queen mating.

A surprising finding is that queens exposed to semen were more likely to get lost than queens that were sham inseminated (subsection “Mating flight behaviour after experimental seminal fluid exposure”, Discussion section). I would have thought that if a queen was so adversely affected by semen that her vision was impaired, she would stay in her hive and lay eggs. 65% of 11 semen-inseminated queens were lost in this experiment (Figure 5). My experience is that only 5% of inseminated queens do not survive to lay eggs. The difference is that standard procedure for instrumentally inseminated queens is to confine them to their hives until they lay. This point should be discussed.

Honeybee queens are highly polyandrous and there is solid experimental evidence that their colonies benefit from increasing levels of genetic diversity in workers (Mattila and Seeley 2007; Mattila et al., 2012). Queens also need to ensure they are fully inseminated and fertile, otherwise they can be killed and replaced by workers. Consequently, queens face trade-offs to ensure they are fully inseminated and that they optimize worker genetic diversity in their hives on one hand and fitness costs incurred by participating in additional mating flights on the other hand. Even if the mortality rate due to male manipulation would be naturally high, the manipulation would still determine an average net increase in reproductive success for first inseminating males if most surviving queens normally obtain a lower number of copulations than the ones they would obtain without any sexual conflict. Once more, we are fully aware that our study establishes the seminal fluid effect on visual perception and mating flight behaviour and offers a parsimonious sexual conflict hypothesis, but we agree that further causation studies will be needed.

The following two papers, which came to diametrically opposed conclusions) are relevant to this manuscript and should be discussed: Schlüns et al., 2005; Tarpy and Page, 2000.

We now cite these studies in the Discussion section and added some text to compare findings of these studies in relation to our results: “The interpretation of our results as being consistent with an ongoing sexual arms race over the number of mating flights rather than the number of copulations *per se* also agrees with previous research suggesting that honeybee queens adjust their flight number based on their insemination success during (a) previous flight(s) (Schlüns et al., 2005). These studies already suggested that natural selection should act primarily at the level of queen flights, which represent greater efforts and risks than individual copulations occurring in quick succession do (Schlüns et al., 2005; Tarpy and Page, 2000). The physiological and/or mechanical mechanisms mediating these responses remain poorly understood and the conceptual logic of our present study provides a novel framework and clear incentive for unravelling them."